# Toward Re-Identifying Any Animal

**Bingliang Jiao**[1,2,3,5]    **Lingqiao Liu**[4]    **Liying Gao**[1,2,3]    **Ruiqi Wu**[1,2,3]
**Guosheng Lin**[5†]    **Peng Wang**[1,2,3†]    **Yanning Zhang**[1,2,3]
[1]School of Computer Science, Northwestern Polytechnical University, China
[2]Ningbo Institute, Northwestern Polytechnical University, China
[3]National Engineering Laboratory for Integrated Aero-Space-Ground-Ocean, China
[4]The University of Adelaide, Australia    [5]Nanyang Technological University, Singapore

## Abstract

The current state of re-identification (ReID) models poses limitations to their applicability in the open world, as they are primarily designed and trained for specific categories like person or vehicle. In light of the importance of ReID technology for tracking wildlife populations and migration patterns, we propose a new task called "Re-identify Any Animal in the Wild" (ReID-AW). This task aims to develop a ReID model capable of handling any unseen wildlife category it encounters. To address this challenge, we have created a comprehensive dataset called Wildlife-71, which includes ReID data from 71 different wildlife categories. This dataset is the first of its kind to encompass multiple object categories in the realm of ReID. Furthermore, we have developed a universal re-identification model named UniReID specifically for the ReID-AW task. To enhance the model's adaptability to the target category, we employ a dynamic prompting mechanism using category-specific visual prompts. These prompts are generated based on knowledge gained from a set of pre-given images within the target category. Additionally, we leverage explicit semantic knowledge derived from the large-scale pre-trained language model, GPT-4. This allows UniReID to focus on regions that are particularly useful for distinguishing individuals within the target category. Extensive experiments have demonstrated the remarkable generalization capability of our UniReID model. It showcases promising performance in handling arbitrary wildlife categories, offering significant advancements in the field of ReID for wildlife conservation and research purposes. Our work is available in `https://github.com/JiaoBL1234/wildlife`.

## 1   Introduction

Object Re-identification (ReID) aims to match identical target objects across non-overlapping camera views. Although existing ReID methods [7, 22, 24, 31, 33] have made the remarkable process, their identification capabilities heavily rely on their training benchmarks. This reliance often results in overfitting to base categories in the training set, *e.g.*, vehicle [22, 33] and person [7, 24, 31]. However, in open-world scenarios, the categories of target objects could be diverse. For example, ReID technologies are widely used toward animal objects for the purposes of analyzing population counts and migration patterns. To adapt these ReID methods to new animal categories, users have to compile large-scale datasets first. This process is time-consuming and poses a significant hurdle to the open-world applications of ReID methods. This limitation motivates us to propose a novel and practical task named "Re-identify Any Animal in the Wild", which aims to construct a universal

---

†*Corresponding author,*
*bingliang.jiao@mail.nwpu.edu.cn, peng.wang@nwpu.edu.cn*

37th Conference on Neural Information Processing Systems (NeurIPS 2023).

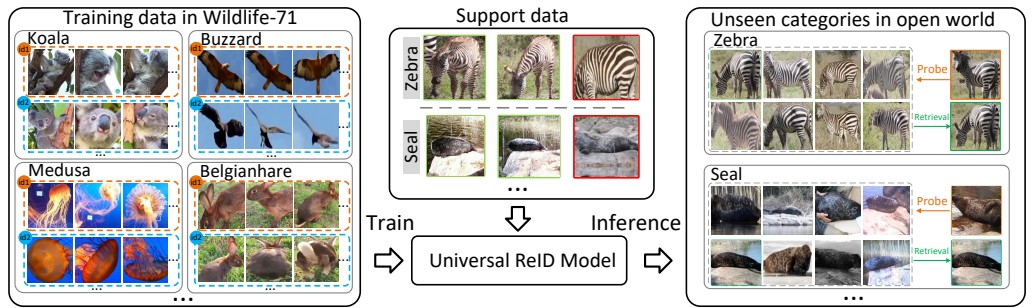

Figure 1: The sketch of the ReID-AW task, which aims to train a ReID model able to handle any animal category under the guidance of a triplet of images within the target category.

ReID model to break the category boundaries and handle unseen animal categories in the open world, as shown in Figure 1.

An effective universal ReID model must fulfill two essential criteria. Firstly, it should have sufficient knowledge learned from an extensive range of seen categories. Secondly, the model should possess the capability to transfer valuable knowledge from seen categories for distinguishing individuals within novel categories. The first criterion necessitates a training dataset that covers diverse object categories, multiple viewpoints, and abundant identities for each category. This ensures that the ReID model can acquire the necessary knowledge for generalization. Unfortunately, existing ReID benchmarks primarily focus on two categories, namely vehicle [22, 33] and person [7, 24, 31], which is insufficient for training a universal ReID model. The second criterion requires the ReID models to be able to adaptively transfer the knowledge learned from the training categories to new categories. It is crucial to recognize that not all the knowledge acquired from the base categories is applicable to the novel ones. For instance, the ReID model might be trained to consider color as a significant attribute for distinguishing many animals, such as dogs and cats. However, such a clue is ineffective or even counterproductive for differentiating chameleons.

In this work, we present the first attempt to construct a universal ReID model. To ensure that the ReID model meets the two aforementioned criteria, we first propose the Wildlife-71 dataset, which contains ReID data from 67 training (base) animal categories and 4 test (novel) categories. In our Wildlife-71 dataset, each training category contains an average of 29 identities, with each identity including approximately 56 images. Our Wildlife-71 dataset is collected from three sources: integrating existing datasets [21, 18, 16, 30, 15], extracting target bounding boxes from a large-scale tracking dataset GOT-10k [9], and crawling web videos to extract target bounding boxes using a tracking algorithm [32]. To the best of our knowledge, Wildlife-71 is the first ReID dataset encompassing multiple object categories. Moreover, in Wildlife-71, we provide a triplet of images (two from the same identity and another from a different identity) as support data for each category in the test set. These support data are pre-accessible for the ReID model to help it adapt to the novel categories. It is worth noting that a triplet image is the smallest unit for learning knowledge to distinguish between individuals, and it is readily available in real-world applications.

In addition to our dataset, we construct a novel universal ReID framework named UniReID. One of the major challenges we faced is equipping UniReID with the ability to select and apply appropriate knowledge for adapting to unseen categories. This step is non-trivial, as it requires the ReID model to both learn the knowledge about the target categories promptly and possess sufficient adaptability to the obtained knowledge. To accomplish this, we train our UniReID to efficiently acquire knowledge about target categories from dual-modal guidance, *i.e.*, visual and textual guidance, and use the knowledge to dynamically adapt to various categories. For visual guidance, we use a triplet of images within the target category to guide the adaption of our UniReID. Particularly, for each training category, we randomly sample an image triplet in training set as guidance, and for test categories, we adopt the given support data. Inspired by visual prompt tuning [10], which efficiently adapts deep models into novel tasks by injecting task-relevant prompt vectors, we employ a transformer layer to transfer the knowledge from the support images into a group of learnable vectors. These vectors then act as visual prompts, adapting our backbone model to the target categories. In addition to the implicit guidance from these visual prompts, we also derive semantic knowledge from a large-scale language model as explicit textual guidance. In this step, we construct a natural language template in the form of "Please give me 4 short phrases describing visual clues beneficial for distinguishing [cls] individuals", where [cls] is the category label, to derive knowledge from the GPT-4 model [19]

in a question-answering manner. GPT-4 is a large-scale pre-trained language model with extensive knowledge and remarkable zero-shot question-answering capability. By this means, we could obtain textual descriptions of crucial clues for distinguishing individuals within the target category. Then, we utilize text-guided attention to help our model focus on these clues. As far as we know, this is the first work introducing knowledge from large-scale language models into ReID tasks. Leveraging such dual-modal guidance, our model could rapidly adapt to unseen categories in the open world.

## 2   Related Work

**Tasks Related to Our ReID-AW**: **1)** *Re-identification.* Re-identification (ReID) aims to match the same object across different camera views. In recent years, numerous studies [24, 6, 31, 27, 22, 14] have been proposed and made significant progress. However, most existing ReID works are specifically designed for a specific category, such as person [6, 27] and vehicle [22, 14], which limits their generalization capability in real-world applications. For instance, many part-based methods [27, 6] assume that pedestrian instances always stand vertically, and they consequently partition pedestrian images into several horizontal stripes to extract local features from head to feet. Although these methods enhance pedestrian re-identification performance, they fail to handle other object categories, such as reptiles, that do not meet the vertically standing assumption. To address this limitation, we propose the ReID-AW task, which aims to design universal re-identification models capable of handling unseen wildlife categories.

**2)** *Domain Generalizable Re-identification.* To facilitate the application of ReID models, Song *et al.* [26] propose the domain generalizable ReID (DGReID) task, which aims to train a ReID model that can work well on unseen environments (domains). Many recent works [26, 12, 4] have been dedicated to addressing the influence caused by style divergence, *e.g.*, viewpoint, illumination, and season variation, across domains to help ReID models generalize to unseen domains. However, DGReID models only consider style divergence and environment variation but fail to account for changes in the target category, which still limits their application in the open world.

**3)** *Few-shot Classification.* Few-shot classification [25, 17] aims to train a robust classifier capable of achieving sufficient performance with a limited amount of support (labeled) data. Notably, there are two major differences between our ReID-AW task and few-shot classification. First, classification focuses on category-level distinctions, *e.g.*, distinguishing between elephant and mouse, where inter-class divergence is considerably large. In contrast, in our ReID-AW, we aim to identify instances within the same category where inter-class divergence can be very subtle. This necessitates ReID models to dynamically adapt to target categories and capture their unique discriminative clues. Second, classification models need only to classify input images into a limited number of categories provided by the support data. As a result, many few-shot classification models [25, 17, 1] can achieve classification by directly calculating the feature similarity between support data and input images without specifically understanding these categories. In contrast, ReID models need to learn sufficient knowledge about the target category to support image matching across unlimited unseen identities.

**Large-Scale Pre-trained Models.** Recently, many works [19, 3, 23] have focused on leveraging abundant training data to endow deep models with comprehensive knowledge and robust representation capabilities. These models have shown their superiority in various vision-language tasks, such as zero-shot classification [23] and visual question answering [19]. For example, CLIP [23] is a lightweight vision-language model trained on 400 million image-text pairs using a contrastive learning strategy, achieving promising zero-shot classification performance. Furthermore, GPT-4 [19] is a robust NLP model trained with 300 billion words, providing it with comprehensive knowledge and remarkable question-answering capabilities. In this work, we aim to derive knowledge from these pre-trained models to facilitate the adaptation of ReID models toward novel categories.

**Prompt Tuning.** As large-scale pre-trained models have shown their superiority in many tasks, numerous works [3, 10, 13, 2] focus on transferring these models to downstream tasks. For example, Brown *et al.* [3] propose to use manually designed natural language templates to derive knowledge from the pre-trained language model, GPT-3, and adapt it to different tasks, such as translation and question-answering. Similarly, Jia *et al.* [10] propose to use a group of task-specific visual prompts to adapt pre-trained vision models to the downstream tasks. In this work, we go one step further to construct category-specific prompts to adapt ReID models to various target categories dynamically.

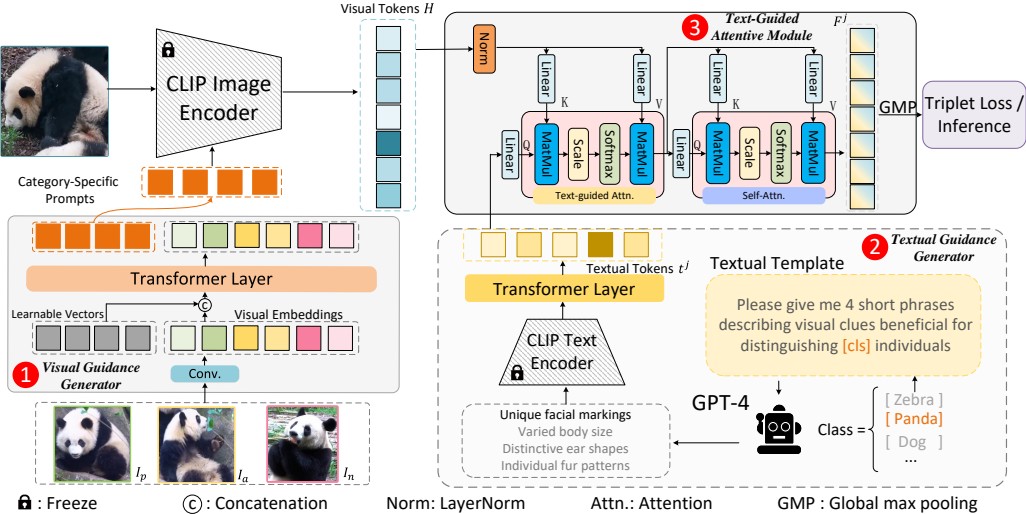

Figure 2: The sketch of our UniReID model. We use pre-trained CLIP as the backbone model. In our UniReID, the visual guidance generator is responsible for utilizing a triplet of images to construct category-specific visual prompts. The textual guidance generator is designed to derive semantic knowledge about the target category from GPT-4 model. The obtained semantic knowledge is then used to guide our UniReID focus on salient visual content via the text-guided attentive module.

## 3 Approach

### 3.1 Problem Definition

In this work, we propose a novel task named "Re-identify Any Animal in the Wild" (ReID-AW). This task seeks to develop a universal ReID model capable of re-identifying animals from unseen classes $C^u$ after being trained on multiple seen classes $C^s$ in our Wildlife-71 dataset. In ReID-AW, ReID models are trained with the training data $\mathbf{D^t} = \{D^k\}_{k=1}^{C^s}$. $D^k = \{(x_i, y_i)\}_{i=1}^{N_k}$ is the labeled training data of the $k$-th category, where $N_k$ is the number of training images in this category. After being trained on $\mathbf{D^t}$, the ReID model is expected to effectively identify individuals belonging to arbitrary unseen class $C^u$. Notably, $C^u$ and $C^s$ is non-overlapped, termed $C^u \cap C^s = \varnothing$. The statistics of our Wildlife-71 are given in **Section 1 in the Appendix**. In summary, Wildlife-71 includes 67 training (seen) categories and 4 test (unseen) categories. To help ReID models adapt to unseen categories, we provide a triplet of labeled images for each test category, serving as support data. The image triplet is denoted as $\{I_a, I_p, I_n\}$, where $I_a$ and $I_p$ are images of the same identity, distinct from $I_n$. ReID models are permitted to pre-access these support data for adaptation before commencing inference on the target categories.

### 3.2 Overall Framework

The overall framework of our UniReID is depicted in Figure 2. As illustrated in this figure, our UniReID model receives dual-modal guidance, *i.e.*, visual and textual guidance, to facilitate adaptation to the target category. The visual guidance consists of a triplet of images, from which our model could learn to identify and capture visual patterns that differentiate target individuals. For textual guidance, we directly derive explicit semantic knowledge from GPT-4 model to inform our model about the discriminative clues of the target category. There are three major components in our UniReID: **1)** *visual guidance generator*, designed to transfer knowledge from a triplet of images into a group of category-specific visual prompts, which is used to adapt our model to the target category; **2)** *textual guidance generator*, employed to derive semantic knowledge about the target category from the large-scale language model GPT-4; **3)** *text-guided attentive module*, which uses the obtained semantic knowledge to adapt our model to the target category through an attention mechanism. These three modules will be introduced sequentially in the following subsections. Additionally, in our UniReID, we utilize the pre-trained CLIP [23] model as the backbone network for extracting features of both input images and textual guidance. This design is based on two key benefits. Firstly, CLIP

is extensively trained on a large number of images containing various object categories, thereby endowing it with robust pattern recognition capabilities that benefit generalization. Secondly, CLIP is trained on an image-text matching task, enabling its visual and textual encoders to naturally embed image and text inputs into a shared space, which greatly facilitates textual guidance over image features.

**Category-specific Sampling Strategy.** During the training stage, we employ a category-specific sampling strategy. Specifically, in each mini-batch, we only sample instances from a single randomly selected category for training. The rationale behind this strategy is that emphasizing cross-category distinguishing could guide ReID models to concentrate more on coarse-grained clues while ignoring fine-grained clues that are actually beneficial for intra-category identification.

## 3.3 Visual Guidance Generator: Category-specific Visual Prompts

The first component of UniReID is the visual guidance generator, designed to use a triplet of images to help our model adapt to target categories. In this step, we find that visual prompt tuning [10] could be an ideal solution, which could efficiently adapt deep models to the target task by introducing task-relevant prompt vectors. Inspired by this, we develop a set of category-specific visual prompts to help our model dynamically adapt to various target categories. Specifically, in our UniReID, we keep all the parameters of the CLIP model frozen (except for normalization layers) and inject it with a set of category-specific visual prompts, which are dynamically constructed based on a triplet of images. Here, for each training category, the image triplet is randomly sampled from the training set, and for test categories, we adopt the support data. As depicted in Figure 2 (1), in this module, we employ a transformer layer to transfer knowledge from the image triplet into a set of learnable vectors, which then serve as category-specific visual prompts. This design has two primary benefits. First, by keeping the parameters unchanged, we can prevent our model from overfitting to seen categories. Second, through the integration of category-specific visual prompts, we can transfer essential knowledge to our model, allowing it to adapt to target categories dynamically.

In practice, given the image triple of target category $\{I_a, I_p, I_n\}$, we first send them into a convolutional encoder to extract their features $\{f_a, f_p, f_n\}$ with $f_* \in \mathbb{R}^{C,H,W}$. Here, $C$ indicates channel dimension, and $(H, W)$ represents spatial scale. Thereafter, we spatially flatten these features and concatenate them together in the flattened spatial dimension. The concatenated features are denoted as $f_{tri.} \in \mathbb{R}^{C,(H \times W \times 3)}$, which is then used to construct category-specific visual prompts. In this work, we adopt deep prompt tuning [10]. This method requires us to construct a visual prompt vector for each layer of the CLIP's visual encoder, *i.e.*, ViT [5]. To do so, we use a transformer layer to transfer the knowledge from $f_{tri.}$ into a group of learnable vectors $L = \{l^1, l^2, ..., l^n\}$, where $l^i \in \mathbb{R}^{C,20}$ is the prepared prompt vector for the $i$-th transformer layer, and $n$ is the depth of visual encoder. This process could be written as,

$$[p^i, \_] = \text{Trans}([l^i, f_{tri.}]), \tag{1}$$

where $p^i$ is the output visual prompts; $\text{Trans}$ is the transformer layer; $\_$ is the output of $f_{tri.}$, which is discarded; $[,]$ indicates concatenation operation.

By this means, we could obtain category-specific visual prompts $P = \{p^1, p^2, ..., p^n\}$, where $p^i \in \mathbb{R}^{C,20}$ is the prompt vector for $i$-th layer of visual encoder. Then, we follow deep prompt tuning [10] to insert $P$ into the CLIP's visual encoder to adapt it to the current category. Formally, we represent the input features of the $i$-th layer of encoder as $\{g^i, h_1^i, h_2^i, ..., h_N^i\}$, where $g^i$ is the class token, $H^i = \{h_1^i, h_2^i, ..., h_N^i\}$ denote hidden features, and $N$ is the length of image tokens. In this step, we first insert $p^i$ into the above feature sequence as $\{g^i, p^i, h_1^i, h_2^i, ..., h_N^i\}$. Then we send them into the $i$-th transformer layer of the visual encoder, which could be written as,

$$[g^{i+1}, \_, H^{i+1}] = \text{Layer}^i([g^i, p^i, H^i]), \tag{2}$$

where $\text{Layer}^i$ is the $i$-th layer of visual encoder; $\_$ is the output of $p^i$, which is discarded. By this means, we could adapt our model toward the target category.

## 3.4 Textual Guidance Generator

In addition to employing implicit guidance from visual prompts, we acquire explicit textual knowledge from pre-trained language models to adapt our model to the target categories. To achieve this, we

design a textual guidance generator to derive explicit textual descriptions toward distinguishable clues of the target category from the GPT-4 model [19]. The GPT-4 is a large-scale pre-trained language model that possesses a wealth of semantic knowledge and boasts impressive zero-shot question-answering performance.

In this part, we follow the approach described in [3] to utilize a natural language template to extract knowledge from the GPT-4 model. Our template is constructed as "Please give me 4 short phrases describing visual clues beneficial for distinguishing [cls] individuals," where [cls] represents the class label of target categories, such as dog, cat, and panda. By populating this template and submitting it to the GPT-4 model, we obtain four textual phrases that outline visual clues useful for distinguishing individuals within the target category, as illustrated in Figure 2 (2). Once we obtain these phrases, we feed them into the textual encoder of the pre-trained CLIP model to extract their features. Notably, to preserve the knowledge of CLIP, we also keep the parameters of its textual encoder frozen. We then pass the extracted features through a trainable transformer layer to adapt them to the current task. The final output textual features are denoted as $T = \{t^1, t^2, t^3, t^4\}$, where $t^j \in \mathbb{R}^{M,C}$ represents the features of the $j$-th phrase and $M$ is the sequence length. These textual features $T$ are then employed as guidance to help our model focus on salient visual content through an attention-based module.

### 3.5 Text-guided Attentive Module

In this subsection, we aim to use the acquired textual features $T = \{t^1, t^2, t^3, t^4\}$ as guidance to assist our model in focusing on visual content that is beneficial in distinguishing individuals within the target category. To accomplish this, we employ a text-guided attentive module to capture salient content over visual features. Formally, given the textual features $t^j$ of the $j$-th textual phrase and patch-level image features $H = \{h_1, h_2, ..., h_N\}$ extracted from CLIP, we first apply linear projection to both of these features as,

$$Q^j = \text{FC}^Q(t^j), \quad K = \text{FC}^K(H), \quad V = \text{FC}^V(H), \tag{3}$$

where $\text{FC}^Q$, $\text{FC}^K$, and $\text{FC}^V$ are fully connected layers used to embed textual and visual features; $Q^j$, $K$, and $V$ represent Query, Key, and Value. Then we use these vectors to calculate attention as,

$$\text{Affinity:} \ S^j = \text{Softmax}(\frac{Q^j K^{\text{T}}}{\sqrt{d}}),$$

$$\text{Aggregation:} \ F_t^j = S^j \cdot V, \tag{4}$$

where $d$ is the channel dimension of $K$; $F_t^j$ is the enhanced feature under the guidance of the $j$-th textual phrase. Particularly, in this work, we follow the multi-head attention [28], partitioning the $Q/K/V$ in the channel dimension into multiple groups and applying attention to each group independently according to Equation 4. Through this attention operation, we could use explicit textual guidance to help our model to focus on distinguishable visual clues. For intuitive examples of how textual guidance aids our model in capturing discriminative clues, please refer to **Figure 2 in the Appendix**. Thereafter, we replay Equation 3 and Equation 4 to apply a self-attention over the $F_t^j$, in which all the $Q$, $K$, and $V$ are embedded from $F_t^j$. Finally, we use a global max pooling to extract the final feature $F^j \in \mathbb{R}^C$, which is used for loss computation and inference.

**Training and Test Procedure.** During the training stage, we use these 4 phrase features to respectively guide the refinement of visual features and utilize refined visual features to calculate triplet loss independently. The objective function could be written as,

$$L = \sum_{j=1}^{4} L_{tri}(F^j, y), \tag{5}$$

where $L$ is the overall objective function of our UniReID; $L_{tri}$ indicates triplet loss; $y$ is the ground truth. During the test phase, we concatenate these 4 refined visual features together for inference.

## 4 Experiments

### 4.1 Datasets and Implementation Details

**Datasets.** *Wildlife-71*. In this work, we evaluate the generalization capability of our UniReID model on our constructed Wildlife-71 dataset. Briefly, our Wildlife-71 consists of 67 training wildlife

Table 1: Comparison results of our UniReID and other ReID methods under the ReID-AW setting. "ReID" and "DGReID" indicate models designed for general and domain generalization ReID settings.

| Type | Method | Reference | Zebra | | Seal | | Giraffe | | Tiger | | AVG | |
|---|---|---|---|---|---|---|---|---|---|---|---|---|
| | | | mAP | CMC-1 | mAP | CMC-1 | mAP | CMC-1 | mAP | CMC-1 | mAP | CMC-1 |
| ReID | TransReID [6] | ICCV2021 | 26.3 | 45.3 | 19.7 | 50.3 | 42.2 | 67.1 | 62.8 | 62.0 | 37.8 | 56.2 |
| | CAL [24] | ICCV2021 | 26.8 | 45.0 | 21.6 | 52.7 | 44.0 | 69.3 | 64.1 | 63.4 | 39.1 | 57.6 |
| | AGW [31] | TPAMI2021 | 24.0 | 40.1 | 23.8 | 55.9 | 42.7 | 68.6 | 58.5 | 59.5 | 37.3 | 56.0 |
| DGReID | IBN-Net [20] | ECCV2018 | 26.2 | 42.6 | 25.2 | 58.0 | 44.0 | 67.5 | 60.5 | 61.5 | 39.0 | 57.4 |
| | SNR [12] | CVPR2020 | 25.5 | 42.4 | 24.1 | 55.0 | 42.5 | 67.0 | 61.5 | 63.4 | 38.4 | 57.0 |
| | MetaBIN [4] | CVPR2021 | 26.4 | 45.6 | 23.9 | 56.7 | 42.6 | 69.3 | 61.2 | 62.0 | 38.5 | 58.4 |
| | DTIN-Net [11] | ECCV2022 | 26.0 | 43.2 | 25.2 | 57.3 | 43.5 | 69.6 | 61.9 | 61.8 | 39.2 | 58.0 |
| ReID-AW | UniReID | – | $29.6^{0.1}$ | $48.7^{0.4}$ | $28.5^{0.2}$ | $62.6^{0.4}$ | $46.5^{0.2}$ | $73.9^{0.7}$ | $66.7^{0.8}$ | $65.2^{1.4}$ | 42.8 | 62.6 |

categories and 4 test wildlife categories, namely zebra [21], seal [18], giraffe [21], and tiger [15]. Each training category contains an average of 29 identities, with each identity including approximately 56 images. To further enrich the training data, we include a person ReID benchmark MSMT17 [30] and a vehicle ReID benchmark VehicleID [16] into the training set of Wildlife-71 as two additional object categories. During the evaluation phase, we evaluate our model on the four test (unseen) categories. For more details about Wildlife-71, please refer to **Section 1 in the Appendix**.

**Implementation Details.** We employ the pre-trained CLIP [23] as the backbone model. During the training and test stage, we resize all images into $224 \times 224$. In the training phase, we adopt the category-specific sampling strategy, as mentioned in Section 3.2. Our model is trained for 20 epochs. The learning rate is initialized to $1.0 \times 10^{-4}$ and divided by 10 at the 15th epochs. Random flipping, random erasing, and color jittering are employed for data augmentation. Two NVIDIA TITAN GPUs are used for model training. We employ the CMC [29] and mAP [34] metrics for evaluation. All ablation experiments in this section are repeated times, and the $\text{mean}^{\text{std}}$ results are reported.

## 4.2 Comparison between Our UniReID with Other State-of-the-art Algorithms

For fair comparisons, all methods (exclude our UniReID) compared on our Wildlife-71 are trained on base categories and then fine-tuned with the support data (a triplet of images) of each test category for 10 iterations.

**Comparison on Wildlife-71.** To evaluate the category generalization capability of our UniReID model, we compare it with other state-of-the-art ReID models on our Wildlife-71 dataset. In this part, we compare our UniReID with models designed for general ReID setting (ReID) and domain generalization ReID setting (DGReID), both of which concentrate on single-category object identification. Specifically, ReID aims to develop effective models that achieve optimal identification performance in a fixed environment, while DGReID aims to design robust models that perform well in unseen environments. The results are presented in Table 1, revealing two key findings. Firstly, on our Wildlife-71 dataset, DGReID methods only marginally outperform general ReID methods, with an average improvement of $1.1\%$ in CMC-1. This observation suggests that re-identifying unseen object categories may be more complex than the DGReID task, requiring ReID models to adaptively accommodate the unique characteristics of novel categories rather than solely addressing style divergence across domains [12]. Secondly, we can find that our UniReID model considerably surpasses all compared methods, achieving an average $3.6\%$ better performance in mAP than the best competitor, DTIN-Net [11]. This outcome implies that the dual-modal guidance employed in our model effectively enables dynamic adaptation to novel categories, leading to enhanced generalization capabilities.

## 4.3 Ablation Studies

To clarify, in the rest of this paper, the "Baseline" indicates a CLIP model trained with visual prompt tuning [10] using static prompt vectors shared for all categories.

**The Effectiveness of Our Designed Modules.** To investigate the effectiveness of our proposed modules, *i.e.*, the category-specific visual prompts and the text-guided attentive module, we incrementally incorporate them into the "Baseline" model and compare the performance improvements. The results, as presented in Table 2, show that both of our designed modules effectively help the ReID model to handle unseen categories. In comparison to the "Baseline" model, our category-specific visual prompts ("Ada.Prompt") result in an average improvement of $1.1\%$ in mAP. This enhancement sug-

Table 2: Ablation experiments of the designed components. The "Visual Prompt Mode" indicates which kind of prompt vectors are used in the current model. "Static" and "Adaptive" represent static prompts shared for all categories and our category-specific prompts, respectively. We can find that both of our designed modules, namely category-specific prompts ("Ada.Prompt") and text-guided attentive module ("Textual Attention"), effectively help the ReID model to handle unseen categories.

| | Visual Prompt Mode | Textual Attention | Zebra | | Seal | | Giraffe | | Tiger | | AVG | |
|---|---|---|---|---|---|---|---|---|---|---|---|---|
| | | | mAP | CMC-1 | mAP | CMC-1 | mAP | CMC-1 | mAP | CMC-1 | mAP | CMC-1 |
| Baseline | Static | × | $26.6^{0.5}$ | $43.8^{0.9}$ | $26.2^{0.4}$ | $58.2^{0.4}$ | $43.8^{0.2}$ | $68.4^{1.2}$ | $62.1^{0.1}$ | $60.3^{0.4}$ | 39.7 | 57.7 |
| Ada.Prompt | Adaptive | × | $27.3^{0.3}$ | $45.4^{0.3}$ | $26.9^{0.7}$ | $59.7^{0.4}$ | $45.6^{0.2}$ | $70.4^{0.7}$ | $63.5^{0.3}$ | $60.7^{1.4}$ | 40.8 | 59.1 |
| UniReID | Adaptive | ✓ | $29.6^{0.1}$ | $48.7^{0.4}$ | $28.5^{0.2}$ | $62.6^{0.4}$ | $46.5^{0.2}$ | $73.9^{0.7}$ | $66.7^{0.8}$ | $65.2^{1.4}$ | 42.8 | 62.6 |

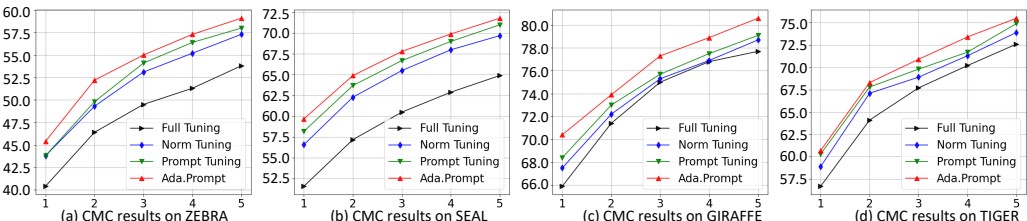

Figure 3: Ablation of our adaptive category-specific prompts ("Ada.Prompt"). CMC-1 to CMC-5 results on 4 test categories are given. We can find that our category-specific prompts tuning strategy outperforms all compared fine-tuning approaches.

gests that these adaptive visual prompts are indeed capable of extracting knowledge from support data and tailoring our UniReID to the target category. Moreover, the text-guided attentive module leads to a further increase of $2.0\%$ in mAP on average. This indicates that using explicit semantic guidance from large-scale language models could indeed help ReID models to capture the discriminative clues of novel categories.

**The Effectiveness of Adaptive Prompt Tuning.** In our UniReID, we use a set of category-specific prompts to adapt our model to novel categories dynamically. To evaluate the effectiveness of our adaptive prompt tuning strategy ("Ada.Prompt"), we compare it with other fine-tuning approaches, namely "Full Tuning" that fine-tunes all parameters of CLIP, "Norm Tuning" [8] that fine-tunes only normalization layers inside CLIP, and "Prompt Tuning" that introduces a set of static prompts [10] shared for all categories on the basis of "Norm Tuning". The comparison results are given in Figure 3, from which we can get three important findings. Firstly, the "Full Tuning" approach demonstrates the poorest performance among all strategies compared. This could be attributed to the fact that modifying the original parameters of CLIP leads to overfitting the base categories, thereby failing to generalize. Secondly, the performances of "Norm Tuning" and "Prompt Tuning" are roughly comparable. These results are expected, as the prompt vectors used in "Prompt Tuning" are especially learned for the base categories, potentially limiting its generalization capabilities towards novel categories. Thirdly, it is evident that our implemented "Ada.Prompt" outperforms all other compared approaches. This could indicate that our adaptive category-specific prompts indeed learn and transfer crucial knowledge to our model, enabling dynamic adaptation to target categories.

**The Effectiveness of Textual Guidance.** From the results in Table 2, we can find that our text-guided attentive module plays a pivotal role, contributing a significant improvement of $2.0\%$ in mAP on average. However, a question may arise: does the model actually learn to utilize the textual guidance, or is this improvement simply the consequence of introducing attention modules? In response to this question, we evaluate the effectiveness of textual guidance from two aspects. **1)** Firstly, we replace the textual tokens in our text-guided attentive module with a set of learnable vectors, thereby converting it into a self-attention module. The results, presented in Table 3, reveal that the self-attention module ("Self-Attn") only brings a $1.0\%$ CMC-1 improvement to the model that does not use attention modules ("w/o Attn"). This could be attributed to the learnable vectors inside the self-attention module potentially overfitting the base categories, leading to poor generalization. This finding indicates that the superiority of our model does not solely stem from the additional attention module. **2)** Secondly, we replace the textual guidance used in our text-guided attentive module with mismatched ones ("Mismatched"), *e.g.*, providing the zebra with monkey guidance, or with a general one ("General") in the format of "Clue one, Clue two, ...". Notably, for the mismatched guidance, we randomly select guidance from three different mismatched categories and averaged the

Table 3: Ablation experiments of textual guidance. The "Self-Attn" is the version that replaces our text-guided attentive module with a self-attention module. The "Mismatched" indicates the version replaces the textual guidance used in our text-guided attentive module with mismatched ones, *e.g.*, providing the zebra with monkey guidance. The "General" is the version that replaces the textual guidance with a group of general phrases like "Salient clue one, ...". We can find that our UniReID with matched guidance ("Matched") outperforms all variations, which indicates our model could make good use of semantic knowledge to adapt to novel categories.

| | Textual Guidance Mode | Zebra | | Seal | | Giraffe | | Tiger | | AVG | |
|---|---|---|---|---|---|---|---|---|---|---|---|
| | | mAP | CMC-1 | mAP | CMC-1 | mAP | CMC-1 | mAP | CMC-1 | mAP | CMC-1 |
| w/o Attn | − | $27.3^{0.3}$ | $45.4^{0.3}$ | $26.9^{0.7}$ | $59.7^{0.4}$ | $45.6^{0.2}$ | $70.4^{0.7}$ | $63.5^{0.3}$ | $60.7^{1.4}$ | 40.8 | 59.1 |
| Self-Attn | − | $28.7^{0.1}$ | $47.0^{0.1}$ | $27.4^{0.2}$ | $60.2^{0.4}$ | $45.5^{0.1}$ | $70.5^{0.2}$ | $65.2^{0.2}$ | $62.6^{0.3}$ | 41.7 | 60.1 |
| | Mismatched | $29.1^{0.2}$ | $47.4^{0.5}$ | $27.8^{0.1}$ | $61.2^{0.8}$ | $46.0^{0.1}$ | $72.6^{0.4}$ | $65.7^{0.3}$ | $64.3^{0.2}$ | 42.2 | 61.4 |
| UniReID | General | $29.2^{0.2}$ | $47.6^{0.1}$ | $28.2^{0.4}$ | $61.4^{0.7}$ | $46.2^{0.1}$ | $73.1^{0.1}$ | $65.9^{0.4}$ | $64.3^{0.9}$ | 42.4 | 61.6 |
| | Matched | $29.6^{0.1}$ | $48.7^{0.4}$ | $28.5^{0.2}$ | $62.6^{0.4}$ | $46.5^{0.2}$ | $73.9^{0.7}$ | $66.7^{0.8}$ | $65.2^{1.4}$ | 42.8 | 62.6 |

Table 4: Comparison results of our UniReID model trained on different datasets, namely vehicle ReID dataset VehicleID, person ReID dataset MSMT17, and our Wildlife-71 dataset. The results indicate that our Wildlife-71 dataset significantly enhances the performance compared to single-category benchmarks.

| Training Set | Zebra | | Seal | | Giraffe | | Tiger | | AVG | |
|---|---|---|---|---|---|---|---|---|---|---|
| | mAP | CMC-1 | mAP | CMC-1 | mAP | CMC-1 | mAP | CMC-1 | mAP | CMC-1 |
| MSMT17 (M) [30] | $23.7^{0.5}$ | $40.5^{1.3}$ | $19.8^{1.5}$ | $45.9^{1.8}$ | $39.1^{1.4}$ | $65.7^{0.4}$ | $45.9^{0.3}$ | $48.2^{0.3}$ | 32.1 | 50.1 |
| VehicleID (V) [16] | $21.1^{1.3}$ | $35.3^{1.7}$ | $23.7^{1.2}$ | $55.6^{1.9}$ | $36.4^{0.2}$ | $59.5^{0.1}$ | $44.6^{0.9}$ | $47.0^{0.3}$ | 31.5 | 49.4 |
| M&V | $24.0^{0.4}$ | $39.9^{0.8}$ | $21.8^{0.7}$ | $51.4^{0.1}$ | $39.5^{1.2}$ | $63.4^{0.9}$ | $47.4^{0.4}$ | $49.9^{0.3}$ | 33.2 | 51.2 |
| Wildlife-71 | $29.6^{0.1}$ | $48.7^{0.4}$ | $28.5^{0.2}$ | $62.6^{0.4}$ | $46.5^{0.2}$ | $73.9^{0.7}$ | $66.7^{0.8}$ | $65.2^{1.4}$ | 42.8 | 62.6 |

final results. The results, presented in the "Mismatched" and "General" rows of Table 3, show that unsuitable textual guidance causes substantial performance degradation compared to using matched ones ("Matched"), *i.e.*, 1.1% CMC-1 on average. This finding indicates that our model indeed learns to utilize semantic knowledge effectively. For intuitive cases of how textual guidance aids our model in capturing salient clues, please refer to **Figure 2 in the Appendix**.

**Effect of Our Wildlife-71 Dataset.** In this work, we construct the Wildlife-71 dataset, which comprises 71 wildlife categories. To the best of our knowledge, our Wildlife-71 dataset is the first ReID benchmark that encompasses multiple object categories. To evaluate the value of this diverse dataset, we compare the performance of our UniReID trained on Wildlife-71 with versions trained on two mainstream ReID benchmarks, namely VehicleID [16] and MSMT17 [30], which only contain pedestrian or vehicle instances. As depicted in Table 4, the performance of the UniReID model trained on the Wildlife-71 dataset surpasses the models trained on single-category benchmarks by a significant margin, achieving an average 12.8% CMC-1 improvement. Furthermore, compared to the version trained on both VehicleID and MSMT17, the UniReID model trained on Wildlife-71 also displays superior performance, with an increase of 11.4% in CMC-1. These results indicate that the diverse categories and the plentiful images in the Wildlife-71 dataset are indeed effective in endowing ReID models with sufficient knowledge for generalization. Therefore, we believe that our Wildlife-71 dataset is a right step toward constructing universal wildlife ReID models for open-world applications.

## 5   Conclusion

In this work, we propose a ReID-AW task, which aims to construct universal ReID models to handle unseen animal categories in the open world. To support the ReID-AW task, we construct a Wildlife-71 dataset, which contains ReID data from 71 wildlife categories. To the best of our knowledge, Wildlife-71 is the first ReID dataset encompassing multiple object categories. Furthermore, we also design a UniReID model, which utilizes dual-modal guidance to dynamically adapt to novel categories. We hope our proposed task, dataset, and approach could support future research on category generalizable ReID methods.

**Limitation.** As pioneers in designing category-generalizable ReID models, our Wildlife-71 dataset only includes limited wildlife categories. In our future work, we will continually enhance our Wildlife-71 dataset and incorporate other categories and more instances.

**Acknowledgements** This work was supported by National Key R&D Program of China (No.2020AAA0106900), the National Natural Science Foundation of China (No.U19B2037), Shaanxi Provincial Key R&D Program (No.2021KWZ-03), and Natural Science Basic Research Program of Shaanxi (No.2021JCW-03).

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
