# Appendix for Toward Re-Identifying Any Animal

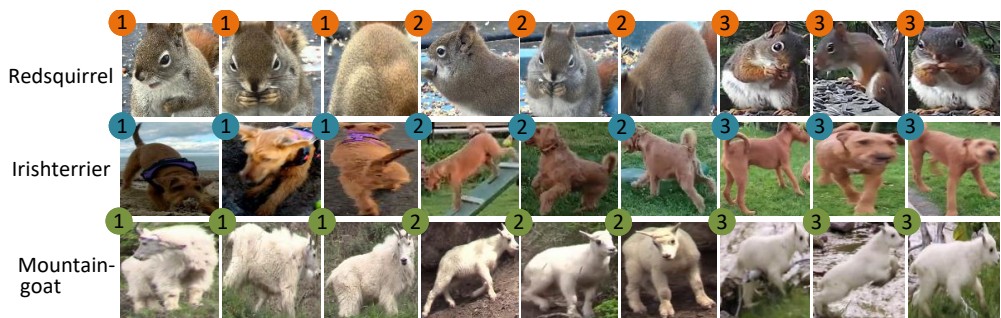

Figure 1: Cases of our Wildlife-71. Images in adjacent three columns belonging to the same identities. We can find that our Wildlife-71 has diverse object categories, various backgrounds, and numerous identities.

## 1  Wildlife-71 Dataset

In this section, we give details about our Wildlife-71 datasets, including the data collection, dataset partition, comparison with other datasets, and future works.

### 1.1  Data Collection

Our Wildlife-71 dataset is mainly collected from three sources, namely integrating existing datasets [8, 7, 3], extracting target bounding boxes from a large-scale tracking dataset GOT-10k [1], and crawling web videos to extract target bounding boxes using a tracking algorithm [10]. Specifically, we incorporate four existing animal ReID datasets as test data into our Wildlife-71 dataset, namely zebra [8], seal [7], giraffe [8], and tiger [3]. Additionally, we gather data from the GOT-10k tracking dataset, which includes over 10k different categories of objects, each category equipped with multiple tracking videos and trajectory annotations for each individual within the videos. In this step, we choose wildlife categories and extract their bounding boxes from videos using the provided annotations. Each trajectory obtained during this process is treated as an individual. We then remove categories with fewer than 10 individuals, leaving us with $1,016$ training identities from 67 different wildlife categories. This step require roughly **40 man-hours**. To further augment the training data for these 67 categories, we collect data from the Internet. Using category class labels like "lion" and "redsquirrel" as keywords, we first crawl web videos from YouTube. Then, we manually filter the obtained videos based on the following criteria: 1) high resolution (greater than $1280 \times 720$), 2) significant viewpoint variation, and 3) diverse backgrounds. This stage consumes approximately **150 man-hours**, obtaining 816 videos across the 67 wildlife categories. Using the acquired videos, we then employ a tracking algorithm [10] to extract individual trajectories. However, due to the imperfect of the tracking algorithm and factors such as camera shake, some trajectories are unsuitable. We tackle this problem by manually selecting trajectories with a sufficient number of bounding boxes (over 10) and removing inaccurate bounding boxes. After refining the data, we obtain 908 trajectories, each treated as an individual. This final step require approximately **200 man-hours**. Examples from our Wildlife-71 dataset are presented in Figure 1.

### 1.2  Dataset Partition

The Wildlife-71 dataset is divided into a training set and a test set. The training set contains $108,096$ images from $1,924$ identities spanning 67 distinct wildlife categories. To further supplement the training data, we integrated training data from a person ReID benchmark MSMT17 [9] and a vehicle

Submitted to 37th Conference on Neural Information Processing Systems (NeurIPS 2023). Do not distribute.

Table 1: Statistics of Wildlife-71 dataset.

| Set | Benchmark | #Category | #Identity | #Images |
|---|---|---|---|---|
| Training | Wildlife (ours) | 67 | $1,924$ | $108,096$ |
| | VehicleID [5] | 1 | $13,164$ | $113,346$ |
| | MSMT17 [9] | 1 | $4,101$ | $124,068$ |
| Test | Zebra [8] | 1 | 546 | $2,958$ |
| | Seal [7] | 1 | 57 | $2,080$ |
| | Giraffe [8] | 1 | 109 | 597 |
| | Tiger [3] | 1 | 107 | 1887 |

Table 2: Comparison with other datasets. "Cams" indicates cameras; "Locs" denotes locations; "Sur." means captured under surveillance cameras; "Web." represents collected from webset.

| Datasets | Object | #Category | Scenario | #Cams/Locs | #Identity | #Images | Average Images |
|---|---|---|---|---|---|---|---|
| Market-1501 [11] | Person | 1 | Sur. | 6 | $1,501$ | $32,668$ | 22 |
| DukeMTMC-reID [12] | Person | 1 | Sur. | 8 | $1,812$ | $34,183$ | 19 |
| CUHK03 [4] | Person | 1 | Sur. | 10 | $1,467$ | $14,097$ | 10 |
| MSMT17 [9] | Person | 1 | Sur. | 15 | $4,101$ | $124,068$ | 31 |
| VeRi [6] | Vehicle | 1 | Sur. | 20 | 776 | $49,357$ | 64 |
| VehicleID [5] | Vehicle | 1 | Sur. | $--$ | $26,267$ | $221,763$ | 8 |
| AIfV [2] | Wildlife | 5 | Web. | 5 | 93 | $20,490$ | 220 |
| Wildlife-71 | Wildlife | 71 | Web. | $1,832$ | $2,743$ | $115,618$ | 42 |

31 ReID benchmark VehicleID [5] into the training set of Wildlife-71 as two additional object categories.
32 The test set of Wildlife-71 comprises four existing wildlife datasets: zebra [8], seal [7], giraffe [8],
33 and tiger [3]. Detailed statistics for our Wildlife-71 dataset are provided in Table 1. Particularly, the
34 original division of the tiger dataset [3] offers only one gallery image per identity. This setup does
35 not align with practical application scenarios, and the limited test set size could lead to large error
36 margins. Consequently, we modified the tiger dataset by integrating its training data into the gallery
37 set.

## 1.3 Comparison with other datasets

39 In Table 2, we compare our Wildlife-71 dataset with existing re-identification datasets across several
40 dimensions, including object type, number of categories, scenario, number of capturing locations,
41 number of identities, the total number of images, and the average number of images per identity.
42 Specifically, we have not incorporated our Wildlife-71 with MSMT17 [9] and VehicleID [5], in
43 this comparison. From this comparison, we observe that in terms of the number of identities, our
44 dataset surpasses most existing benchmarks, with the exception of VehicleID [5] and MSMT17 [9].
45 Additionally, each identity in our Wildlife-71 dataset contains over 42 images on average, supassing
46 those of VehicleID (8 images per identity) and MSMT17 (31 images per identity). Moreover, as
47 our Wildlife-71 dataset was compiled from web videos, it boasts a significantly larger number
48 of capturing locations than other datasets, which are gathered through fixed surveillance cameras.
49 Besides, compared with the existing animal dataset AIfV [2], our Wildlife-71 contains significantly
50 more categories, identities, and images. Particularly, considering the limited identities and images,
51 the AIfV is constructed only for the evaluation purpose rather than training a category-generalizable
52 wildlife re-identification model. For other existing wildlife datasets like Zebra [8], Seal [7], Giraffe [8],
53 and Tiger [3], we have included them into our testing set, the information of which is given in Table 1.

## 1.4 Future work and extension version.

55 Esteemed peer reviewers provided valuable recommendations to enlarge the dataset, thereby am-
56 plifying its practical value. Taking this in mind, we keep continually expanding our Wildlife-71
57 dataset. As of now, the wildlife categories have grown to 106, and the count of wildlife identities in
58 the training set has been expanded about 10 times (about 19000). Moving forward, we will polish
59 the collected data while continuing its expansion. The extended version will be released soon, and
60 we remain committed to continually refining and expanding our Wildlife-71 dataset in subsequent
61 research endeavors.

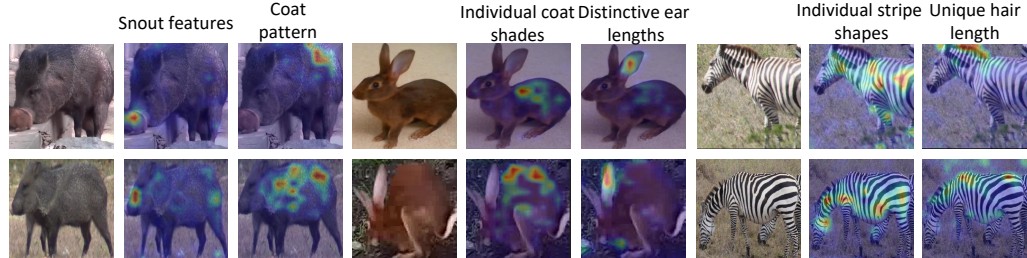

Figure 2: Visualization of activation maps generated by our text-guided attentive module. The first row is the employed textual guidance and the next two rows are corresponding activation maps. We can find that our text-guided attentive module could indeed make good use of the textual guidance and help our model focus on discriminative clues of target categories.