# OpenReview forum: "Toward Re-Identifying Any Animal"
_NeurIPS.cc/2023/Conference — NeurIPS 2023 poster_

### Official Review · Reviewer_ABAL · 2023-06-13

**Soundness:** 2 fair
**Presentation:** 3 good
**Contribution:** 2 fair
**Rating:** 2
**Confidence:** 5

**Summary:**

The paper focuses on the ReID problem. Different from previous works, which mainly focus on persons and vehicles, the paper wants to reID any animals. To realize this, the authors construct a new dataset that contains many animals. Also, the authors propose a visual guidance generator, a textual guidance generator, and a Text-Guided attentive module. By integrating the power of CLIP and GPT4, authors claim that they achieve re-identifying "ANY" animals.

20230808 Update: after checking the cons by other reviewers, I still lean to reject the paper. Kindly ask the fellow reviewers check what I concern, thanks.

**Strengths:**

Personally, I appreciate the paper and spent a whole day on this paper as well as its supplementary. What makes me interested in this paper is the following:

(1) The interesting idea: Re-Identifying Any Animal is very interesting and promising to me. It has more influence than Re-Identifying people and vehicles. I think it is a step to Re-Identifying Anything.

(2) The good writing: I can fully understand what authors want to express by reading only once. The writing is smooth and excellent.


**Weaknesses:**

Although the paper is very interesting and the topic authors discussed is very promising, I feel there are some major concerns. Below please find my comments on these concerns. Please correct me if I’m wrong.

(1) The very small dataset. Building a dataset is very admirable in the data-centric AI era. However, I think the significance of the dataset is limited. As you claim in the title, you want to reID any animals; however, in the Sup Table 1 and 2, there are only 2671 identities in your proposed dataset. The number of identities is even much smaller than Vehicle datasets (Vehicle) and Person datasets (MSMT17). In the ReID task, each identity is regarded as one class. To achieve "ANY", you should include a very large identity, for example, 267100. As expected, with such a small number, you can only freeze the CLIP backbone and use the GPT4 API (I see the fine-tuning results in worse performance). In conclusion, I doubt the value of the proposed Wildlife-71. What I expect is very large re-identifying datasets and the methods are ONLY trained on this to achieve "ANY". So here, my suggestion is to enlarge the dataset by 100 times to see what happens.

(2) The proposed method lacks novelty. I understand that it is acceptable that the main contribution is a dataset, and the proposed method is just regarded as a strong baseline to benchmark the proposed dataset. However, as I stated in (1), I think the value of the dataset is limited. The method part is built on existing works: 1. The CLIP and VPT: you only regard the supported images as prompt to tune efficiently. It is not novel and easy to do. 2. The Text-Guided Attentive module: there are so many methods [1-3] used to aggregate the information from vision and languages. May I know what new insights are considering these prior works? So here, my suggestion is only to focus on the training method on your proposed dataset. The model pre-training on a huge dataset (assuming you have labeled one) is very attractive. I see you only have two TITAN GPUs; using at least 8*A100 is more suitable for the "ANY" task. Apologize in advance if you do not have this.

(3) The insufficient and not convincing experiments. I know the paper contains a braving idea, however, I find: 1. With the help of the strongest GPT4 and CLIP, your method only archives about +3% over other famous ReID and DGReID methods (Table 1). In my opinion, the improvement is too marginal, even with the strongest tools. 2. I would suggest adding comparisons with existing aggregation modules [4-6], which is lacking in this paper. 3. The design of the visual guidance generator: I see that you compare the tuning method in your proposed methods and others. What I am concerned about is if there are any other kinds of visual generators. Do they perform well? 4. I am curious about how to select the supported images in the test process. Do you assume there are labels? Do you use any matching methods? Comparisons are needed. 5. During the test, you concatenate the 4 visual features. That incurs how much computing burden. Please show the inference speed compared with other methods. If the added computational cost is significant compared with the accuracy gain, it is hard to judge the advantage of this system. If the authors could add the suggested experiments it would be very good.

[1] Coarse-to-Fine Vision-Language Pre-training with Fusion in the Backbone.
[2] Language As Queries for Referring Video Object Segmentation.
[3] mPLUG: Effective and Efficient Vision-Language Learning by Cross-modal Skip-connections.
[4] Attend-and-Excite: Attention-Based Semantic Guidance for Text-to-Image Diffusion Models
[5] Non-local Neural Networks (you may adapt it to utilize text)
[6] A Text Attention Network for Spatial Deformation Robust Scene Text Image Super-resolution


**Questions:**

Some small questions:

1. What is the meaning of  "20" in Line 199?

2. How many prompts are used in each ViT layer?

3. Remove the Duke dataset due to its invasion of privacy.

4. If you want to work on reID anything, how can you obtain the dataset? Have you considered the synthetic dataset?

5. Do you think it is possible to reID any animals without any help from the tools like GPT4? I am charmed at Vision Foundation (Big) Models. I want you to focus on this.

---

> ### Author Rebuttal · Authors · 2023-08-09
>
> 1.Data:
> For the open-world problem, there are two mainstream solutions. One is using huge dataset to train a strong deep model, such as SAM. The other one is transfer-learning methods, e.g., UniDetetcor [A]. These methods employ a dataset with a certain volume to gather knowledge and increase the models' adaptability to adapt the accumulated knowledge to handle novel instances they encountered.
>
> In this study, we chose the second, data-efficient, solution, since collecting a huge wildlife dataset could hardly be achieved in the foreseeable future. Since ReID dataset requires multiple frames for each instance, its collection could not be achieved by simply crawling web images as SAM. So, we collect wildlife ReID data from videos. But, due to the rarity of wildlife, we found that for each category, only an average of 50 available videos could be collected on Youtube. Thus, gathering your excepted dataset requires laborers to identify, track, and photograph over 260K animals in the wild. Given the limited economic value of wildlife, in the foreseeable future, there could hardly be any institution to afford such costs.
>
> Although Wildlife-71 may be not large enough to train a foundation model, its value should not be overlooked. As shown in Table 4, compared with existing datasets, wildlife-71 could provide ReID models with effective knowledge to identify animals. Besides, incorporating the textual knowledge derived from GPT-4 and our proposed category adaptation modules, the knowledge inside wildlife-71 could be further adapted to unseen animals, which indicates the value of wildlife-71 to support the second solution.
>
> Finally, our objective is toward (not achieved) re-identifying any animal. If you feel it's over-claimed, we are happy to replace it, e.g., "Toward category-generalizable wildlife re-identification".
>
> 2.Novelty:
> In terms of the overall design, our UniReID offers two pivotal insights. The first is deriving textual guidance from LLMs into the ReID model, empowering it with adequate knowledge for generalization. As R@9xKB also notes, this solution could inspire future research using LLMs. The second is empowering ReID models with sufficient adaptability to adapt accumulated knowledge to handle unseen categories. These insights could benefit both ReID-AW and analogous cross-category instance retrieval scenarios.
>
> For technical details, **1)** Our visual prompt is not a tuning strategy. Unlike VPT methods that need to train a set of prompt vectors for each target task, our visual guidance generator (ViGG) could generate category-specific prompts through a fully forward process. As far as we know, this is the first work to construct visual prompts in such an adaptive and tuning-free manner. **2)** Rather than revise the attention module, our major consideration during incorporating textual prompts is to address the inherent modality gap between them and images and ensure they can accurately associate with visual cues without explicit semantic annotations (e.g. segmentation maps). For this issue, we propose to use the CLIP model, adeptly trained to bridge visual and textual elements, embedding them into a shared hidden space for the subsequent attention operation.
>
>
> 3.Performance:
> GPT-4 and CLIP are large language and vision-language models. They suffer significant modal gaps with fine-grained visual tasks. Hence, their application to ReID-AW is not straightforward and does not necessitate improvement. As shown in Figure 3, a simple application of CLIP to ReID-AW results in unsatisfactory results. To address this, we propose the visual and textual prompting strategy, which helps to bridge this gap and make use of the knowledge in these models. The effort involved in this process should not be overlooked.
>
> 4.Aggregation:
> We replace our textual attentive module with these three aggregation modules and evaluate them on ReID-AW. [4] achieves 60.4% CMC-1 (v.s. UniReID's 61.4%). [5] replaces the attentive module with a transformer decoder, achieving a comparable performance, i.e., 61.1. [6] introduces a Gaussian filter to capture salient clues comprehensively, which slightly increases the CMC-1 of UniReID by 0.2%.
>
> 5.Generators:
> We replace ViGG with two other hyper-networks, namely, the fusion net[B] and the MLP net[C]. [B] uses 64 learnable vectors as templates and then uses features of support images to predict a set of fusion weights to aggregate these templates into visual prompts. [C] receives the features of support images and adaptively embeds them into visual prompts via an MLP. For ViGG, we were devoted to integrating the benefits of both methods above. In ViGG, we use a set of trainable vectors to accumulate knowledge and a transformer layer to calibrate them into category-specific prompts under the guidance of support images. Hence, UniReID suppresses the version using the above two methods by 1.0% and 0.8% CMC-1.
>
> 6.Support images:
> For each test set, we randomly select a triplet of images and designate them as the support data, which, once designated, remains consistent across all experiments.
>
> 7.Speed:
> Please refer to Answer 2 for R@9xKB.
>
> 8.20:
> 20 is the number of prompts used in each ViT layer.
>
> 9.Duke:
> We would remove experiments on Duke.
>
> 10.Synthetic:
> Yes. Although models like Diffusion can spawn diverse instances, they struggle to produce images of the same identity across varying viewpoints and poses. Recently, we discern that advancements in 3D generation methods may offer a solution. Such methods can create an array of 3D models, thus securing inter-identity diversity. 3D models can also be manipulated to generate images of the same identities.
>
> 11.Tool:
> Since collecting a huge wildlife dataset could hardly be achieved in the near future, tools like GPT-4 are valuable.
>
> [A] Detecting everything in the open world: Towards universal object detection CVPR23\
> [B] Dynamic convolution: Attention over convolution kernels CVPR20\
> [C] Decoupled dynamic filter networks CVPR21

---

> > ### Comment · Reviewer_ABAL · 2023-08-10
> > **1: Data**
> >
> > **Authors rebuttal**
> >
> > 1.Data: For the open-world problem, there are two mainstream solutions. One is using huge dataset to train a strong deep model, such as SAM. The other one is transfer-learning methods, e.g., UniDetetcor [A]. These methods employ a dataset with a certain volume to gather knowledge and increase the models' adaptability to adapt the accumulated knowledge to handle novel instances they encountered.
> >
> > In this study, we chose the second, data-efficient, solution, since collecting a huge wildlife dataset could hardly be achieved in the foreseeable future. Since ReID dataset requires multiple frames for each instance, its collection could not be achieved by simply crawling web images as SAM. So, we collect wildlife ReID data from videos. But, due to the rarity of wildlife, we found that for each category, only an average of 50 available videos could be collected on Youtube. Thus, gathering your excepted dataset requires laborers to identify, track, and photograph over 260K animals in the wild. Given the limited economic value of wildlife, in the foreseeable future, there could hardly be any institution to afford such costs.
> >
> > Although Wildlife-71 may be not large enough to train a foundation model, its value should not be overlooked. As shown in Table 4, compared with existing datasets, wildlife-71 could provide ReID models with effective knowledge to identify animals. Besides, incorporating the textual knowledge derived from GPT-4 and our proposed category adaptation modules, the knowledge inside wildlife-71 could be further adapted to unseen animals, which indicates the value of wildlife-71 to support the second solution.
> >
> > Finally, our objective is toward (not achieved) re-identifying any animal. If you feel it's over-claimed, we are happy to replace it, e.g., "Toward category-generalizable wildlife re-identification".
> >
> >
> > **My opinion**
> > 1. The paper UniDetetcor you mentioned also uses a very large dataset (*These contributions allow UniDetector to detect over 7k categories, the largest measurable category size so far, with only about 500 classes participating in training.*). Admittedly it does not train on a very large dataset, the test categories/data are much larger than yours and therefore have more meaning. For the current version, I do not see your methods have experimented on such big dataset. Therefore, I prefer a limited contribution.
> >
> > 2. Admittedly, collecting such a big dataset is very expensive, I do not think it is impossible. For example, the collection of dataset OpenImage by Google is much harder than the dataset I suggested.
> >
> > 3. Although you want to under-claim, the performance is too low. I do not think it has any practical use.
> >
> > 4. For the synthetic data, I mean, use some engine such as Unity to build. For example, please refer to the RandPerson and ClonePerson.
> >
> > **In conclusion, the rebuttal in the data view does not satisfy me. Suggest a Reject in the aspect of data.**
> >
> > I am not sure the authors will see this because:
> > **Error: NeurIPS 2023 Conference Submission3282 Authors must not be readers of the comment**

---

> > > ### Author Response · Authors · 2023-08-11
> > > **Response to the concern of R@ABAL about Data**
> > >
> > > While we believe that accumulating huge data to train a large model is not the only factor in solving open-world problems, however, your views on this issue seem firm.
> > >
> > > From our opinion:
> > > 1. As you notice, UniDetetcor is also an annotation-efficient model, which uses only images belonging to 500 categories to train a generalizable model. This indicates that our transfer-learning solution is also valuable and feasible. For UniDetetcor ‘s 7,000 test category,  it is attributed to the fact that detection is not a task as fine-grained as ReID, and detection already owns many existing multi-category datasets like VisualGenome for evaluation. In contrast, for our Wildlife-71 dataset, even facing the scarcity of wildlife data and the fine-grained nature of ReID task, we have gathered data from 7 categories (4 listed in the paper and 3 newly added) , which could also assess our model's generalization. This effort should not be regarded as a limitation of this work.
> > >
> > > 2. **Firstly**, the volume of data isn't the sole metric for evaluating a dataset's value, and our Wildlife-71 encompasses over 100K images, larger than most ReID benchmarks (Table 1 in Appendix).
> > > **Secondly**, given the limited economic value of wildlife, in the foreseeable future, there could hardly be any institution to afford the costs to build a wildlife ReID dataset of the same volume as OpenImage. However, we contend that research should not be evaluated solely on its economic merits. The social merits of wildlife re-identification technology, especially in areas like endangered population monitoring and migration pattern analysis, holds substantial value, driving our thorough research.
> > >
> > > 3. **Firstly**, open-world problem is a challenging task that needs to be solved gradually and collaboratively by community researchers, rather than addressed in one step. For instance, our admitted UniDetetcor also only brings limited improvement (3.8 AP in its Table 4) to previous works.  In comparison, our UniReID brings 4.5 CMC-1 improvement to the top-performing generalizable ReID model DTIN-Net.
> > > **Secondly**,  as the pioneer ``toward'' (we did not claim we have achieved)  re-identifying any animal, we introduce the significant ReID-AW task, construct a benchmark encompassing over 100,000 images (outstripping the size of most existing ReID datasets), establish an evaluation protocol, and give our technical solution. We posit that our efforts could introduce this essential task to the re-identification community, which should not be fully negated based solely on dataset volume considerations. Meanwhile, we believe that with the collaborative efforts of the community, both the Wild-71 dataset and the ReID-AW task will achieve notable progress.
> > >
> > > 4. Notably, to produce synthetic data for each wildlife category using tools like Unity, we would have to gather tens of thousands of 3D scans for each wildlife category to create a 3D parametric model similar to SMPL. This process could arguably be more time-consuming than collecting ReID data itself. Consequently, we suggest leveraging nerf-based 3D generation models like [1] as we discussed in response, which could leverage 2D data that is comparatively easier to amass. However, given the imperfection of these methods, they could not be directly employed yet.
> > >
> > >
> > > [1] Wang Z, et al. “ProlificDreamer: High-Fidelity and Diverse Text-to-3D Generation with Variational Score Distillation.”, arxiv.

---

> > > > ### Comment · Reviewer_ABAL · 2023-08-11
> > > >
> > > > I need to clarify that I am not firmly resisting accumulating huge amounts of data as the only way to achieve in-context learning. But the thing is currently with small data and borrowing power from pre-trained models have very low accuracy (See tables in your paper). Therefore, focusing on data is more useful and reasonable to achieve in-context learning.
> > > >
> > > > Suggestions: you can gain a better understanding in the workshop which also focuses on the data: https://sites.google.com/view/vdu-cvpr22: **This workshop aims to bring together research works and discussions focusing on analyzing vision datasets, as opposed to the commonly seen algorithm-centric counterparts.** I cannot agree more with the opinions in that workshop.

---

> > > > > ### Author Response · Authors · 2023-08-11
> > > > > **Response to the concern of R@ABAL about Data:**
> > > > >
> > > > > As you mentioned, relying solely on training large models isn't the only approach to address the open-world problem, and transfer learning could also be effective when we can hardly collect sufficient data for training foundation models, such as in the case of wildlife. With that in mind, we kindly request that you evaluate our dataset on its value for transfer learning and adaptation, rather than focus solely on the consideration of its volume.

---

> > ### Comment · Reviewer_ABAL · 2023-08-10
> > **2.Novelty**
> >
> > **Authors rebuttal**
> >
> > 2.Novelty: In terms of the overall design, our UniReID offers two pivotal insights. The first is deriving textual guidance from LLMs into the ReID model, empowering it with adequate knowledge for generalization. As R@9xKB also notes, this solution could inspire future research using LLMs. The second is empowering ReID models with sufficient adaptability to adapt accumulated knowledge to handle unseen categories. These insights could benefit both ReID-AW and analogous cross-category instance retrieval scenarios.
> >
> > For technical details, 1) Our visual prompt is not a tuning strategy. Unlike VPT methods that need to train a set of prompt vectors for each target task, our visual guidance generator (ViGG) could generate category-specific prompts through a fully forward process. As far as we know, this is the first work to construct visual prompts in such an adaptive and tuning-free manner. 2) Rather than revise the attention module, our major consideration during incorporating textual prompts is to address the inherent modality gap between them and images and ensure they can accurately associate with visual cues without explicit semantic annotations (e.g. segmentation maps). For this issue, we propose to use the CLIP model, adeptly trained to bridge visual and textual elements, embedding them into a shared hidden space for the subsequent attention operation.
> >
> > **My opinion**
> >
> > Different people see the novelty from different views. I acknowledge the contribution: (1) prompts an in-context learning setting (2) the aggregation of text and image. But they are not convincing enough for me.
> >
> > I recommend a **Weak Reject** here for the novelty aspect.
> >
> > I am not sure the authors will see this because: **Error: NeurIPS 2023 Conference Submission3282 Authors must not be readers of the comment**

---

> > > ### Author Response · Authors · 2023-08-11
> > > **Response to the concern of R@ABAL about Novelty**
> > >
> > > I'm sorry, but your statement "Different people see the novelty from different views. " is very confusing and somehow subjective. As you admitted, the utilization of category-adaptive features and the incorporation of text-guided representations to integrate pre-existing category-related knowledge from large language models inside our model is novel in ReID community and could be extended to analogous cross-category instance-level retrieval scenarios. Meanwhile, from our perspective, these two points above potentially provide more technical insights for future research than directly using huge data.

---

> > > > ### Comment · Reviewer_ABAL · 2023-08-11
> > > >
> > > > Acknowledged with the novelty you argue. But still think it is marginal/incremental and below the NeuIPS's bar.

---

> > ### Comment · Reviewer_ABAL · 2023-08-10
> > **3.Performance**
> >
> > **Authors rebuttal**
> >
> > 3.Performance: GPT-4 and CLIP are large language and vision-language models. They suffer significant modal gaps with fine-grained visual tasks. Hence, their application to ReID-AW is not straightforward and does not necessitate improvement. As shown in Figure 3, a simple application of CLIP to ReID-AW results in unsatisfactory results. To address this, we propose the visual and textual prompting strategy, which helps to bridge this gap and make use of the knowledge in these models. The effort involved in this process should not be overlooked.
> >
> > **My opinion**
> > The authors do not reply to my comments about poor improvement. I acknowledge that directly using CLIP and GPT-4 has no improvement. But that should not be the reason for you use powerful tools but only gains a poor improvement.
> >
> > **Strong Reject** for the experiment part.
> >
> > I am not sure the authors will see this because: **Error: NeurIPS 2023 Conference Submission3282 Authors must not be readers of the comment**

---

> > > ### Author Response · Authors · 2023-08-11
> > > **Response to the concern of R@ABAL about Performance**
> > >
> > > **Firstly**, as far as we know, there is no theory proving that the use of CLIP and GPT-4 will necessarily lead to amazing improvements in open-world visual tasks. Notably, another work, UniDetetcor, using CLIP to address open-world detection task also only brings limited improvement (3.8 AP in its Table 4).
> > >
> > > **Secondly**, as you notice that, directly using CLIP and GPT-4 has no improvement in ReID-AW. Hence, the efforts we have made to make them work should not be overlooked (Table 2).

---

> > > > ### Comment · Reviewer_ABAL · 2023-08-11
> > > >
> > > > **(1)**  I disagree that *there is no theoretical or empirical proof that the use of CLIP and GPT-4 will necessarily lead to amazing improvements in open-world visual tasks.* Many works applies CLIP to open-world tasks, such as:
> > > > 1. PointCLIP V2: Adapting CLIP for Powerful 3D Open-world Learning
> > > > 2. CLIP-FO3D: Learning Free Open-world 3D Scene Representations from 2D Dense CLIP
> > > > 3. Zero-Shot Out-of-Distribution Detection Based on the Pre-trained Model CLIP
> > > >
> > > > **(2)** In your response, you admit **your method brings limited improvement**, and claim that you should not be accused because other accepted works also did it. I disagree. For example, you should not say you run a red light because others do it without penalty.
> > > >
> > > > **(3)** In addition, CLIP and GPT-4 are powerful zero-shot tools, which have great potential in open-world settings.

---

> > > > > ### Author Response · Authors · 2023-08-11
> > > > > **Response to the concern of R@ABAL about Performance:**
> > > > >
> > > > > First, we aimed to clarify that “as far as we know, no theory proving that the use of CLIP and GPT-4 will necessarily lead to **amazing** improvements.” rather than deny the potential value of them toward the open-world problem. In fact, in this work, we notice this value and aim to exploit it.
> > > > >
> > > > > Second, the purpose we cite UniReID is to illustrate the issues of “necessarily” and “amazing” rather than follow it to ”run the red light“. Notably, the PointCLIP V2 you pointed out, which has made good use of the knowledge inside the clip, also brings comparable improvement with ours (5 R1 list in its Tab. 1).

---

> > ### Comment · Reviewer_ABAL · 2023-08-10
> > **Others**
> >
> > Others are clear. My main concerns lie in **(1) the contributed data (2) the novelty (3) the performance**. I sincerely ask my fellow reviewers and AC to consider this.
> > Thanks

---

> ### Comment · Reviewer_ABAL · 2023-08-10
> **Questions**
>
> Besides the responses to your rebuttal, I am considering what is the practical meaning of re-ID any **ANIMALS**? I cannot imagine a use of your system.

---

> > ### Author Response · Authors · 2023-08-11
> > **Response to the Questions of R@ABAL**
> >
> > "Besides the responses to your rebuttal, I am considering what is the practical meaning of re-ID any ANIMALS? I cannot imagine a use of your system."
> >
> > We highly contend that research should not be evaluated solely on its economic merits. The social merits of wildlife re-identification technology, especially in areas like **endangered population monitoring** and **migration pattern analysis**, holds substantial value, driving our thorough research.

---

> > > ### Comment · Reviewer_ABAL · 2023-08-11
> > >
> > > I agree that social merits are as important as economic merits. But I still think your proposal is far from reaching the value (endangered population monitoring and migration pattern analysis) you argued. Does your benchmark consider the images of wild animals captured in a long time? The look of an animal will be very different during migration, and the reID system will fail. The reID for animals is far different from that for people: animals do not wear clothes which are one of the most discriminative features for humans.

---

> > > > ### Author Response · Authors · 2023-08-11
> > > > **Response to the questions of R@ABAL:**
> > > >
> > > > The social merits like endangered population monitoring and migration pattern analysis are not only the potential practices of the ReID-AW task but also our major motivation to introduce it to the re-identification community. For some potential issues like appearance changing with time, they could be addressed technically rather than solely based on large benchmarks. Specifically,  when the appearance of wildlife varies, users could adjust the provided prompt (images and textual) to adapt our UniReID to the changed patterns and capture discriminative clues accordingly. At the same time, putting aside our discussion of migration analysis, the role of ReID-AW in endangered population monitoring seems like more acceptable.

---

### Official Review · Reviewer_5HdD · 2023-06-19

**Soundness:** 4 excellent
**Presentation:** 3 good
**Contribution:** 3 good
**Rating:** 5
**Confidence:** 4

**Summary:**

This paper introduces a new task called "Re-identify Any Animal in the Wild" (ReID-AW) which aims to develop a ReID model capable of handling any unseen wildlife category it encounters. To address this challenge, the authors created a comprehensive dataset called Wildlife-71, which includes ReID data from 71 different wildlife categories, and developed a universal re-identification model named UniReID specifically for the ReID-AW task. The authors employed a dynamic prompting mechanism using category-specific visual prompts to enhance model adaptability and leverage explicit semantic knowledge derived from GPT-4 to focus on regions useful for distinguishing individuals within the target category. UniReID showcases promising performance in handling arbitrary wildlife categories, offering significant advancements in the field of ReID for wildlife conservation and research purposes.

**Strengths:**

1. this paper presents a new generic task compared to object re-identification and provides a dataset including 71 categories.
2. For this task, the authors propose a generic re-identification model that combines visual and textual features to re-identify categories that were not seen during model training.


**Weaknesses:**

1. The authors only evaluated performance comparisons in categories that had not been seen for the experimental evaluation. However, comparisons in seen categories, such as pedestrian versus other methods, are missing.
2. Some minor issues： the authors do not list the contributions of this paper in an organized manner, and random erasing lacks corresponding citations [1].


**Referecens**
[1] Zhong Z, Zheng L, Kang G, et al. Random erasing data augmentation. Proceedings of the AAAI conference on artificial intelligence. 2020, 34(07): 13001-13008.

**Questions:**

1. The authors indicate that " The rationale behind this strategy is that emphasizing cross-category distinguishing could guide ReID models to concentrate more on coarse-grained clues while ignoring fine-grained clues that are actually beneficial for intra-category identification". The current understanding of intra-class and inter-instance differences in the ReID task is limited due to the previous studies' sole focus on the differences between classes. Hence, it remains unclear how these differences could affect the model's performance. Additionally, while the results indicate better performance on unseen classes, the accuracy of the model on the seen classes is questionable. Why not consider all variations within and between classes to gain a comprehensive understanding of the ReID task and improve the accuracy of the model?

2. The authors give an interesting example in the introduction section using chameleons, why not use this as a test set or whether this dataset contains that category and how it performs in that category?

The authors have proposed an interesting task and I think the dataset is very important for re-identifying communities, however, there are some doubts about the construction of the dataset and the input on the method, I will raise the score if the authors have solved my problem well.



**Limitations:**

This seems to limit the use of the model because the model must use a triplet as the visual input.

---

> ### Author Rebuttal · Authors · 2023-08-09
>
> 1.**Performance comparisons in seen categories.**\
> To address this concern, we evaluated our UniReID model and the top-performing competitors, DTIN-Net [A], on two seen categories: alligator and morgan. Here, we need to clarify that we are continually augmenting our Wildlife-71 dataset and have amassed over 1 K new wildlife identities. These newly acquired data are used to construct the test set above. In these categories, our UniReID achieves 71.2% and 55.8% CMC-1 accuracy, respectively. Meanwhile, the DTIN-Net achieves  67.8% and 52.7%, demonstrating that our UniReID model averagely outperforms DTIN-Net by 3.3%, which indicates the superiority of our model. For a detailed comparison in the context of the person ReID task, please refer to Section 2 in the Appendix.
>
> 2.**the authors do not list the contributions of this paper in an organized manner, and random erasing lacks corresponding citations [1].**\
> Thanks for your suggestions. We would refine our paper to list our contributions and include this citation. Specifically, the contributions would be summarised as follows:
>
> 1) In this research, we propose a novel task named "Re-identify Any Animal in the Wild" (ReID-AW). The objective of ReID-AW is to develop a universal ReID model capable of handling any unseen wildlife category it encounters.
> 2) To equip the universal ReID model with sufficient knowledge for generalization, we construct a diverse dataset named Wildlife-71 for training purposes, which includes ReID data from 71 different wildlife categories. To the best of our knowledge, Wildlife-71 is the first ReID dataset encompassing multiple object categories.
> 3) In addition to the dataset, we also develop a novel ReID framework named UniReID for the ReID-AW task. Within our UniReID, a visual dynamic prompting mechanism and a textual attentive module are given, which are responsible for leveraging visual and textual guidance to adapt our model toward the target category.
>
> 3.**The authors indicate that "The rationale behind this strategy is that ...".**\
> Firstly, in terms of experimental results, removing our category-specific sampling strategy could result in a 2.7% CMC-1 degradation to our final model. The reason behind this might be that the knowledge employed by ReID models to distinguish instances from varied categories can be ineffective, or even detrimental when applied to identifying instances within the same category. Intuitively, if we concentrate the ReID model on differentiating instances between horse and cow during the training stage, it could regard coarse-grained clues such as the presence or absence of horns, as a key distinguishing clue. However, such coarse-grained clues are ineffective for fine-grained intra-category identification among horses. Additionally, it is worth mentioning that this observation has also been made by Jiao et.al. [A]. in the context of the domain generalization person ReID task. They discovered that when training a ReID model to differentiate samples from different domains, the model could be misled into considering coarse-grained cues like backgrounds and seasons as primary factors, which, however, are ineffective for person ReID. Consequently, they introduced a domain-specific sampling method, which aligns in principle with our approach.
>
> 4.**Evaluation on the chameleon category.**\
> In the previously submitted version of Wildlife-71, we have not included the chameleon category. Until now, as we continuously expand our dataset, we have amassed 134 distinct chameleon identities with various colors. Here, we utilized these newly acquired data as a test set and assessed the performance of our UniReID specifically for the chameleon category. In comparison with the top-performing competitor, DTIN-Net [A], our UniReID model demonstrated a 6.6% improvement in CMC-1 performance (68.9% v.s. 62.3%), which indicates the superiority of our model.
>
> [A] Jiao BL, et al. Dynamically transformed instance normalization network for generalizable person re-identification. ECCV, 2022.

---

> > ### Comment · Reviewer_5HdD · 2023-08-11
> > **The author's response addresses most of my concerns and the authors are encouraged to provide some statistics on the final dataset**
> >
> > Upon carefully reviewing the comments from both the reviewers and the authors' responses, my positive evaluation of this paper remains unchanged. The primary factor behind my positive stance is that this paper presents an intriguing task; however, it is possible that the claim of re-identifying any animal in the given topic may be overly ambitious, as pointed out by Reviewer#ABAL.
> >
> > Similar to the concerns expressed by Reviewer#ABAL, my primary concern pertains to the dataset provided by the authors. The authors have encountered challenges regarding the size of the provided dataset; nevertheless, they have included various intriguing categories, including chameleons. Notwithstanding the limited size of the current dataset, the authors are actively expanding its volume. Hence, I kindly request the authors to furnish statistical information regarding the final dataset (the version at the time of the paper's official publication if it is accepted), thereby enabling all reviewers to reevaluate the significance of this study.

---

> > > ### Author Response · Authors · 2023-08-11
> > >
> > > Thanks for your support and admission. We will continue to expand our Wildlife dataset, whether our paper is accepted or not. Besides, we will give the latest statistical information of our dataset as an extended version in our final paper.

---

### Official Review · Reviewer_LLxx · 2023-07-02

**Soundness:** 3 good
**Presentation:** 3 good
**Contribution:** 2 fair
**Rating:** 4
**Confidence:** 5

**Summary:**

This paper proposes a new task called “Re-identify Any Animal in the Wild” (ReID-AW) and created a comprehensive dataset called Wildlife-71 which is used to evaluate ReID-AW methods. Furthermore, the authors present a universal re-identification model named UniReID specifically for the ReID-AW task. This model receives dual-modal guidance, i.e., visual and textual guidance, to facilitate adaptation to the target category. Experimental results show that the UniReID model considerably surpasses all compared methods.

**Strengths:**

1) New task: The authors propose a practical task named “Re-identify Any Animal in the Wild” and first created a comprehensive dataset encompassing multiple object categories.

2) Integration of new technologies: The method uses the large-scale pre-trained model and visual prompt tuning to improve model performance.

3) comprehensive experiment: The method yields good performance under multiple settings.

**Weaknesses:**

**Major Concerns**
1) My major concern is that the proposed framework is a good combination of many popular techniques:

**a) GPT-4; b) multi-modal CLIP; c) Visual prompt tuning [10]; And focuses on a new dataset for d) animal re-identification.**

a) The using of GPT-4.0 seems just generate 4 fixed text phrases. This operation could be simply replaced by any other manual annotation or search engine including wiki.
The use of CLIP in b) and c) generation of visual prompts seems interesting while using the text guidance for visual features is quite common.
For d), there exist many datasets focusing on animals, including tigers, and fishes. Thus why not directly use these datasets considering they are relatively large? And I am still confused why this proposed method only focuses on animal re-identification, but not other objects including vehicles or human beings.

2) Some details are not clear to me. As a Re-identification task, how do the authors generate category-specific prompts for unseen ids? In my view, the unseen IDs during inference do not have learnable specific prompts for further tuning. This design seems unnatural in the re-identification task.

3) There are also some other works using CLIP models for the same task of re-identification. This paper could incorporate it into a comparison or at least a discussion.

[A] *Li, S., Sun, L., & Li, Q. (2023, June). CLIP-ReID: Exploiting Vision-Language Model for Image Re-identification without Concrete Text Labels. In Proceedings of the AAAI Conference on Artificial Intelligence (Vol. 37, No. 1, pp. 1405-1413).*

**Other Minor Concerns**

4) GPT-4 is not an open-source model. So the textual guidance generator relying on GPT-4 may be limited in its use.

5)   The authors claim that“we aim to identify instances within the same category where inter-class divergence can be very subtle”. This statement is not clearly explained and the authors do not discuss this in detail.

**Questions:**

Please refer to the weakness section. The authors could respond to why these techniques are necessary and why it is distinctive for only the animal re-identification task. Or the animal re-ID is just a newly-proposed setting to attract the research attention? I cannot see why this new setting is distinctive and should only be solved by these new techniques.

Due to time limitations, the authors could focus on my major concerns as a matter of priority.

**Limitations:**

The limitation is clearly stated after the conclusion. The authors claimed that they will incorporate other categories in future work, which is also one of  my major concern for this manuscript.

---

> ### Author Rebuttal · Authors · 2023-08-09
>
> 1.**Why techniques are necessary and why it is distinctive for animal**\
> Sorry for causing confusion. Here, we need to clarify that the techniques encompassed within our model are not designed to fit animal objects but address the primary challenge of the ReID-AW task, i.e., identify instances within unseen categories. It is worth noting that existing ReID methods are primarily designed and trained to handle instances within a specific category. For instance, a tiger ReID model [B] could only identify tiger instances. In this work, we attempt to transcend this category boundary and train a universal ReID model on limited seen animal categories that can generalize to unseen animal categories in the wild. To the best of our knowledge, research on such cross-category ReID remains an unexplored domain and presents a considerable challenge. Besides, we need to clarify that while our evaluation primarily centers on animal ReID, the method we present holds inherent potential for extension to a broader array of cross-category instance-level retrieval (CCIR) challenges. The foundational elements established in our approach offer two pivotal insights that can be harnessed for analogous CCIR scenarios: first, the utilization of category-adaptive representations, and second, the incorporation of text-guided representations to integrate pre-existing category-related knowledge from large language models (LLMs).
>
> 2.**Combination of popular techniques**\
> Please note that our model is not merely a naive combination of existing techniques; instead, it is carefully constructed based on our two key insights. Specifically, our first insight is to incorporate textual guidance from LLMs into the ReID model, empowering it with adequate knowledge to facilitate generalization. As R@9xKB also notes, this solution could inspire future research using LLMs. The second insight is to empower the ReID model with sufficient adaptability to efficiently adapt accumulated knowledge to handle unseen target categories. We believe that these two insights could not only address the ReID-AW but also be harnessed for analogous CCIR scenarios. \
> Regarding some specific concerns, we address them below.
>
> 1) GPT-4 and multi-modal CLIP: Leveraging textual knowledge from GPT-4 is non-trivial. Unlike the visual guidance widely used in existing ReID models, e.g., keypoints [B], this textual guidance cannot be directly applied to image features due to the inherent modality gap.
> To alleviate this gap and ensure that textual prompts can accurately associate with visual clues without explicit semantic annotations (e.g. segmentation maps), we propose to employ the CLIP model, adeptly trained to bridge visual and textual elements, embedding them into a shared hidden space for the subsequent attention operation.
>
> 2) Animal ReID dataset: Existing animal ReID datasets focus solely on a few categories. The limited knowledge contained in these datasets is insufficient to support the training of a universal ReID model that could generalize to unseen animal categories. Therefore, we construct a more comprehensive dataset Wildlife-71 which contains 71 different animal categories to train the universal ReID model. For the existing animal datasets, i.e., Zebra, Seal, **Tiger**, and Giraffe, we use them as test sets to evaluate the category generalization capability of ReID models.
>
> 3) Besides, our UniReID model does not overfit the animal objects. As we exhibited in Section 2 in Appendix, our UniReID also achieves state-of-the-art performance under the domain generalization person ReID task.
>
> 3.**Prompts for unseen ids**\
> During the inference phase, when deploying our UniReID model to address a specific target (unseen) category, such as the tiger, we initially provide a triplet of tiger images as support. Acquiring such minimal support data is feasible in real-world scenarios. Leveraging these support images, our visual guidance generator (elaborated in Section 3.3) can produce category-specific visual prompts shared for all target tiger images through a one-time, **fully forward process**. Armed with these category-specific prompts, our UniReID model can efficiently adapt to the tiger category without any training or tuning.
>
> 4.**CLIP-ReID**\
> We compare our UniReID with the CLIP-ReID model [C] under ReID-AW setting. Across the four test sets, our UniReID surpasses the CLIP-ReID model by 6.3% CMC-1 on average (61.4 vs 55.1). The reason could be that the CLIP-ReID focuses on using knowledge embedded within CLIP, but falls short in adapting the accumulated knowledge to novel categories, which could be addressed by the visual and textual prompting strategy of our UniReID.
>
> 5.**Relying on GPT-4**\
> Our UniReID does not rely on the usage of GPT-4. In fact, replacing the GPT-4 to GPT-3, our UniReID still averagely outperforms the best competitor DTIN-Net [A] by 4.1% CMC-1 under ReID-AW.
>
> 6.**Explanation of “we aim to identify...”**\
> This statement is intended to clarify the primary distinction between the few-shot classification (FSC) and our ReID-AW. Specifically, our ReID-AW represents a more fine-grained task compared to FSC. Compared with the FSC that seeks only to determine the category of a target instance (*e.g.*, distinguishing between an elephant and a mouse), our ReID-AW goes one step further, diving into each specific category to recognize the unique identity of a target instance (*e.g.*, identifying one particular elephant from other elephants). As a result, the inter-class (*i.e.*, inter-identity) divergence in ReID-AW is much smaller than the inter-class (*i.e.*, inter-category) divergence in FSC, which inherently makes ReID-AW a more challenging task.
>
> [A] Dynamically transformed instance normalization network for generalizable person re-identification. ECCV22.\
> [B] Part-pose guided amur tiger re-identification. ICCVW19.\
> [C] CLIP-ReID: Exploiting Vision-Language Model for Image Re-identification without Concrete Text Labels. AAAI23.

---

> > ### Author Response · Authors · 2023-08-12
> > **Supplement illustration about our insight behind adopting GPT-4 and designed visual prompts.**
> >
> > Due to space limitations, we did not illustrate our insight behind adopting GPT-4 and our designed visual prompts in Answer 2. Here, we would like to offer supplementary illustrations.
> >
> > 1) **GPT-4**: In fact,  choosing GPT-4 is based on the consideration of its two superiorities. Firstly, GPT-4, as one of the most advanced LLMs, possesses substantial knowledge beneficial for category generalization. Secondly, after the API of GPT-4 being released, our model could derive knowledge on-the-fly in a question-answering manner (as we designed in Section 3.4) without any manual operation.
> >
> >
> > 2) **Visual prompt tuning**: As you kindly note that the visual guidance strategy within our UniReID also represents a significant innovation. Different from existing visual prompt tuning methods [A], which require training a unique set of prompt vectors for each downstream task, our visual guidance generator can produce category-specific prompts via a **fully forward process**. To the best of our knowledge, this is the first attempt to formulate visual prompts in such an adaptive and tuning-free fashion. This innovation enables our UniReID model to efficiently adapt to unseen categories.
> >
> > [A] Jia M, et al. ``Visual prompt tuning'', ECCV, 2022.

---

> > > ### Comment · Reviewer_ABAL · 2023-08-12
> > >
> > > It is not the first: please refer to
> > >
> > > 1. Visual Prompting via Image Inpainting
> > >
> > > 2. Images Speak in Images: A Generalist Painter for In-Context Visual Learning
> > >
> > > 3. Towards In-context Scene Understanding
> > >
> > > 4. Personalize Segment Anything Model with One Shot
> > >
> > > 5. In-Context Learning Unlocked for Diffusion Models
> > >
> > > 6.  Exploring Effective Factors for Improving Visual In-Context Learning
> > >
> > > 7.  What Makes Good Examples for Visual In-Context Learning?

---

> > > > ### Author Response · Authors · 2023-08-12
> > > >
> > > > Thanks for your suggestion and sorry for causing confusion.
> > > >
> > > > Here, we would like to discuss our "interesting generation manner of visual prompts" inside deep visual prompt tuning (DPT) technology [A], which is kindly noticed by **R@LLxx**. Compared with existing works [A,B] adopting DPT technology,  we giving an effective attempt to construct the visual prompt vector inside DPT based on a more efficient generation manner rather than fine-tuning. This innovation allows our model to adapt to the target categories efficiently.
> > > >
> > > > Regarding the idea of utilizing images as guidance for adapting the deep models, as you pointed out, it's a senior approach in existing one-shot tasks. We wouldn't claim to be the pioneers of this idea in our work.
> > > >
> > > > If you are concerned that we are over-claim here, we would like to replace it with **In the ReID domain, we give the first attempt to learn category-adaptive features by producing the visual prompts inside [A] in such an adaptive and tuning-free manner, which allow our model to adapt to the target categories efficiently.**
> > > >
> > > > [A] Jia M, et al. ``Visual prompt tuning'', ECCV, 2022.
> > > > [B] Zhou Z, et al. Zegclip: Towards adapting clip for zero-shot semantic segmentation, CVPR, 2023

---

> > ### Comment · Reviewer_LLxx · 2023-08-15
> > **Reply to rebuttals**
> >
> > I appreciate the authors for their detailed responses and added new results to address my concerns.
> > I carefully read the discussions of other reviewers and the author's response.
> >
> > At this stage of discussion, I still have serious doubts about the novelty of this paper.
> > From my point of view, the use of large language models such as GPT-4 is a highlight of this paper, and it is also a key technology for why this proposed method can recognize "any" animal if I understand correctly.
> > However, by using GPT-4, the proposed technique seems to just change the original category word into a description of an object and input it into the CLIP model. For example, the original word description for "tiger" has been changed to a word description for "some kind of four-legged reptile mammal". At this point, it seems that any linguistic dictionary can be used as a substitute for GPT-4 of any known human thing. Therefore, I wonder what the authors and other reviewers think about this issue. It's hard for me to call it an effective highlight or novelty.  And it is hard to say it is an innovation in machine learning or computer vision.
> >
> > As other reviewers mentioned, one crucial issue for Re-ID tasks is its usability in the realistic world, and I feel that the authors did not fully convince the reviewers of this.

---

> > > ### Comment · Reviewer_ABAL · 2023-08-15
> > >
> > > Thanks for your reply. You point out that **any linguistic dictionary can be used as a substitute for GPT-4 of any known human thing.** I missed that when I reviewed the paper. I agree with you that the GPT-4 is not necessary here though the authors give some comparative experiments. Generally, I think the paper  is **chasing the clout**, and should not be accepted at all.

---

> > > > ### Author Response · Authors · 2023-08-15
> > > >
> > > > Thanks for your suggestion. No matter whether this paper is accepted or not, we believe the comments from all respected reviewers could help us to enhance its quality.
> > > >
> > > > For your concerns about the necessity of LLMs inside our model, please refer to our response for R@LLxx.

---

> > > ### Author Response · Authors · 2023-08-15
> > > **Response to Concerns of R@LLxx**
> > >
> > > First, we would like to extend our sincere gratitude for your careful review and comments.
> > >
> > > 1.**The necessity of LLMs**: We are sorry for causing confusion about the role of LLMs in our approach. For the utilization of textual prompts given by the LLMs, our goal is to equip our UniReID model with the **knowledge about which specific clues are discriminative for identifying individuals within the target category** (like "Unique facial markings" and "distinctive ear shapes" for the panda, as shown in Figure 2), rather than the description of the general characteristics of target category wildlife (like tiger is four-legged reptile mammal). With such fine-grained textual prompts, we could effectively guide our UniReID model to use appropriate knowledge and capture actual discriminative clues (via attention) for identifying target category individuals.
> > >
> > > Besides, it is worth noting that providing such fine-grained knowledge is non-trivial, which needs a systematic understanding and summary of existing knowledge related to target categories and **could hardly obtain straightforwardly from existing linguistic dictionaries**. Meanwhile, considering that ReID-AW is an open-world problem aimed at generalizing to unpredictable and unseen target categories, it is also not feasible to mutually conduct exhaustive summarization and annotation for all potential target categories. To address this issue, in this work, we made an insight that the significantly developed LLMs could offer a promising solution. Specifically, the rich knowledge and reasoning ability of LLMs enables them to summarize their learned knowledge related to target categories into the textual prompts about discriminative clues, which could be used as guidance to adapt ReID models.
> > >
> > > As far as we know, in the ReID domain, our UniReID gives the first attempt to construct category-adaptive and discriminative features to identify individuals of unseen category leveraging knowledge from LLMs, which we believe could provide insights to a broader array of cross-category instance-level retrieval challenges. Meanwhile, as **R@9xKB** mentioned, “This can inspire researchers in using GPT”.
> > >
> > >
> > >
> > > 2.**Usability of ReID-AW**: For the usability of the ReID-AW task, as mentioned by **R@5HdD**, it could be valuable in the field of wildlife conservation. Specifically, a major motivation that drives us to propose the ReID-AW is to address some crucial tasks in protecting endangered wildlife, e.g., endangered population monitoring. For these endangered wildlife categories, a dataset with enough samples to train a specific ReID model could hardly be collected due to their rarity. Based on this observation, we construct our Wildlife-71 dataset and propose the ReID-AW task, which aims to develop a ReID model trained on our collected 67 wildlife categories that could generalize to unseen wildlife categories. We believe such models could be useful for relevant wildlife conservation tasks (as mentioned in our abstract and introduction).
> > >
> > > Finally, we hope that the respected reviewer, when assessing the research and practical value of our ReID-AW, could take into account the positive comments from other reviewers and consider the efforts we have made to introduce it to the community.

---

### Official Review · Reviewer_9xKB · 2023-07-07

**Soundness:** 3 good
**Presentation:** 3 good
**Contribution:** 3 good
**Rating:** 7
**Confidence:** 5

**Summary:**

This paper extends common person or vehicle reid task to re-identifying any animal in the wild, called ReID-AW task. First, authors propose an animal reid dataset for the ReID-AW task that contains 71 different categories. Then, authors also propose a new method for this task to tackle specific challenges in this task. Specially, authors propose to use GPT-4 to generate prompts to guide attention learning on the visual embedding, which can inspire researchers in using GPT.

**Strengths:**

This paper extends common person or vehicle reid task to re-identifying any animal in the wild, called ReID-AW task. First, authors propose an animal reid dataset for the ReID-AW task that contains 71 different categories. Then, authors also propose a new method for this task to tackle specific challenges in this task. Specially, authors propose to use GPT-4 to generate prompts to guide attention learning on the visual embedding, which can inspire researchers in using GPT.

**Weaknesses:**

--- In experiments, only “zebra”, “seal”, “giraffe”, and “tiger” are used as test set. It’s better to evaluate the performance by setting other categories as test sets.
--- Compared with other methods, the proposed method may suffer from the high computation complexity.

**Questions:**

--- What is the difference between using GPT-4 and other LLMs such GPT-3 or llama?
--- If some instances change along with time, will the model work?
--- Authors use 4 test prompts in this method. Will using more prompts works better?
--- Will the model trained on Wildlife-71 work well on person reid dataset?

**Limitations:**

See Weakness

---

> ### Author Rebuttal · Authors · 2023-08-09
>
> 1. **Evaluate the performance on other categories.**\
>    Thank you for your suggestion. To extend our test set, we have further incorporated new instances from two seen categories, *i.e.*, alligator and morgan, as well as from an unseen category named chameleon. In these three categories, our UniReID achieves CMC-1 accuracy of 71.2%, 55.8%, and 68.9%. Compared with the top-performing competitor, DTIN-Net [A], which respectively achieves 67.8%, 52.7%, and 63.3% CMC-1 accuracy, our UniReID markedly outperforms, which indicates its superiority.
> 2. **Comparison of computation complexity with other methods.**\
>     When compared with the TransReID [B] method, which uses the same ViT backbone as our UniReID, the floating-point operations in our model increase only by 1.4 GMACs (19.7 v.s. 18.3). Additionally, the time taken to process and infer a single query image (tested on the Zebra dataset) shows a minor increase of 0.002 seconds (0.044 v.s. 0.046), which we consider to be acceptable. It is important to highlight that for a specific target category, both visual and textual prompts are consistent across all input images of that category. This means their features could be calculated only once and can then be reused for all test images. Apart from this, the additional computational consumption of our UniReID model merely stems from the text-guided attentive module.
> 3. **The difference between using GPT-4 and other LLMs such GPT-3 or llama.**\
>    As one of the most advanced language models,  the GPT-4 is trained with more data than GPT-3 and llama. Nonetheless, it is crucial to note that the superiorities of our UniReID do not depend solely on the usage of GPT-4. In fact, replacing the GPT-4 to GPT-3, our UniReID only suffers average 0.3% mAP degradation (42.5 v.s. 42.2).
> 4. **If some instances change along with time, will the model work?**\
>    Our UniReID possesses the capability to adapt to varying inputs. Crucially, UniReID can adapt to instances from any specific target category via a fully forward process, guided by the provided visual and textual prompts, without any need for fine-tuning. Consequently, if target categories and instances change over time, a mere modification of the visual and textual prompts could adapt our model accordingly. This adaptability is a significant advantage of our UniReID model.
> 5. **Will using more text prompts works better?**\
>    To respond to this issue, we increase the number of textual prompts to 6 and 8, respectively, and utilize them to train our UniReID. From the results, we find that using 6 and 8 prompts could only bring an average 0.3% and 0.2% mAP improvement to our UniReID. Upon analyzing the generated prompts, we observed that compelling GPT-4 to produce an excessive number of prompts could potentially cause them to concentrate on less distinctive clues, which can hardly result in performance improvement.
> 6. **Will the model trained on Wildlife-71 work well on person reid dataset?**\
>    To address this concern, we evaluated our UniReID model, trained on the Wildlife-71 dataset, using the test set of the Market-1501 dataset. Compared to the UniReID model trained under the person ReID dataset (details are given in Section 2 of the Appendix), the model trained on Wildlife-71 shows a 29.2% decrease in CMC-1 performance (72.6 v.s. 43.4). This decline could be attributed to the significant appearance and semantic differences between wildlife and pedestrians. The knowledge accumulated in Wildlife-71, such as patterns of fur, feathers, or horns, could hardly generalize to pedestrian instances within Market-1501.
>
>
> [A] Jiao BL, et al. Dynamically transformed instance normalization network for generalizable person re-identification. ECCV, 2022.\
> [B] He ST, et al. TransReID: Transformer-based object re-identification. ICCV, 2021.

---

### Comment · Reviewer_ABAL · 2023-08-11
**Conclusion**

Many thanks for the authors' replies and discussion. After several discussions, my main concerns remain on **Data, Novelty, and Performance**. Also, a small concern is **whether the proposed task is really meaningful.**

I have also acknowledged the authors' opinion on the aforementioned aspects.

My final decision is **Strong Reject**.

Thanks.

---

> ### Comment · Reviewer_ABAL · 2023-08-11
>
> I hope to listen to the opinion of Reviewer 9xKB and Reviewer LLxx.

---

> ### Author Response · Authors · 2023-08-11
> **Response to the conclusion of R@ABAL**
>
> Firstly, we would like to express our sincere gratitude for all R@ABAL's positive comments. Your suggestions on our experiments and presentations are certainly helpful in enhancing the quality of our paper.
>
> After numerous discussions and considering your final comments, I think your primary concerns are as follows:
>
> 1)**Dataset**: You suggest that the volume of our dataset is a significant limitation since it is insufficient for training a foundation (big) model. However, as we agreed in our discussion, the foundation model trained on huge data **is not the sole solution for open-world problems**. Hence, the value of our dataset, particularly for **another solution** (we adopted) based on transfer learning and adaptation, should not be overlooked. Besides, as **R@5HdD** (positive rating) kindly noticed that while we do encounter challenges in collecting wildlife data, we have included various intriguing categories and **are actively expanding its volume**.
>
> 2)**Novelty**: We are sorry, but we think that the suggested "Different people see the novelty from different views." seems not a convincing reason to reject our research.
>
> 3)**Experiments**: After the response, we are glad to find that you accept our ablation and comparison experiments. The only remaining experimental concern seem that the usage of CLIP and GPT-4 has not brought improvement you expected.  Although we have attempted to illustrate that utilizing CLIP and GPT-4 doesn't "necessarily" lead to "amazing" improvement in open-world visual tasks, it seems you do not agree with our perspective. Of course, we value and appreciate your diverse perspectives in research. However, it's not convincing that you specifically rate our experiment as **Strong Reject based solely on this single concern and perspective divergence**.
>
> Finally, we sincerely hope that the respected reviewers can review our discussion, and recognize our efforts as pioneers in introducing the important ReID-AW task to the community.

---

> > ### Comment · Reviewer_ABAL · 2023-08-11
> >
> > The "Different people see the novelty from different views."  is not a reason for rejecting your paper. It is thought of **limited novelty** that pushes me to reject the paper. Please see the original novelty comments for why I think the limited novelty.

---

> > > ### Author Response · Authors · 2023-08-11
> > >
> > > We sincerely apologize for our misunderstanding about your concern.
> > >
> > > Based on your comment about our Novality: "Different people see the novelty from different views. I acknowledge the contribution: (1) prompts an in-context learning setting (2) the aggregation of text and image. But they are not convincing enough for me.".
> > >
> > > Our insights: (1) the utilization of category-adaptive features (2) the incorporation of text-guided representations to integrate pre-existing category-related knowledge from large language models, seem acknowledged.
> > >
> > > These two insights we believe are novel in the ReID community and could be extended to analogous cross-category instance-level retrieval scenarios. The primary divergence in our perspectives seems to be the novel degree to these insights.

---

### Author Response · Authors · 2023-08-21
**Summarization**

As the discussion phase is coming to its end, we would like to summarize our perspectives for the respected reviewers’ and ACs' convenience.

**Dataset**: To the best of our knowledge, our proposed Wildlife-71 is the first cross-category ReID dataset, encompassing ReID data from 71 diverse wildlife categories. We sincerely hope that the respected reviewers could evaluate it from not only its size (which suppress most existing ReID datasets, as illustrated in Table 1 of the Appendix) but also its potential contributions to transfer learning and adaptation learning research, and its value to the ReID community.

**Approach**: In this work, we developed a UniReID model based on the idea of transfer learning. In this model, we propose visual and textual prompting strategies to guide it to adaptively transfer knowledge acquired from our Wildlife-71 dataset toward unseen wildlife categories, which enables it to identify individuals within these unseen categories. We sincerely hope that the respected reviewers could recognize the critical insights offered by our UniReID model: **1)** Deriving valuable knowledge from LLMs to guide ReID models to efficiently adapt to unseen categories (for the details, please refer to our discussion with R@LLxx), and **2)** Learning category-adaptive features to assist the ReID model in flexibly generalizing to various target categories. We believe these two insights could not only benefit addressing the ReID-AW task but also inspire future research on cross-category instance-level retrieval challenges.

**ReID-AW Task**: To the best of our knowledge, our ReID-AW task gives the first attempt at the cross-category ReID. We sincerely hope that the respected reviewers could recognize its research value as well as its potential contribution to wildlife conservation. Besides, we also hope that when evaluating the contributions of this work, our efforts of introducing this task to the ReID community could be considered.

Finally, we would like to extend our sincere gratitude to all reviewers for their time and effort invested in reviewing this paper. We will take your constructive suggestions and persist in expanding our dataset and improving our work.

---

> ### Comment · Reviewer_ABAL · 2023-08-21
>
> Thanks for authors’ many days hard working and discussing your paper with me. Hope your paper accepted by a later conference. Sorry for insisting my score.

---

> ### Comment · Reviewer_5HdD · 2023-08-22
> **Thanks as well to the author for the reply and the effort put in**
>
> Proposing a new task and building the corresponding dataset takes more effort than simply improving performance on an existing task. I appreciate the work, and I appreciate the effort you guys put in even more. Good luck with your next submission.

---

> > ### Comment · Reviewer_ABAL · 2023-08-22
> >
> > Proposing a new task and building the corresponding dataset may not be as difficult as you think. I have experience with four works (one published, one rejected, one recently done, and one ongoing) for proposing a new task and building the datasets. Personally, I think this kind of work has no need to compare with SOTAs, which saves much effort. It generally took three months to finish one. Beating SOTA (for example, proposing a new method to improve DETR performance on COCO) is very time-consuming and GPU-consuming. Given the strong background (sorry for being confident) on this kind of paper, I recommend rejection.

---

> > > ### Author Response · Authors · 2023-08-22
> > >
> > > Thanks for your suggestion.
> > >
> > > However, we think **took three months to finish one** new task and dataset is somehow not precise. The Open-Image dataset you mentioned took the Google team many years to annotate and refine. It is also worth noting that our dataset contains over 100,000 images, and the process of annotation, organization, and polishing has already taken more than 3 months.
> > >
> > > Besides, we have a different perspective from your suggested **no need to compare with SOTAs**. In fact, in this work, we not only propose a new task but also give an effective framework to address it. This effort should not be overlooked. Hence, in our ReID-AW task, we have also compared with SOTA ReID methods to evaluate our methods.

---

> > > > ### Comment · Reviewer_ABAL · 2023-08-22
> > > >
> > > > Thanks for the reply:
> > > >
> > > > 1. I agree that famous datasets, such as OpenImage, take years to finish. But with enough funds and label resources, labeling your dataset (I have checked the construction process in your supplementary) does not take a long time. Please do not re-emphasize that my focus is the size of the data. I also care about novelty and experiments. The size is only a part of my concern.
> > > >
> > > > 2. Sorry for not stating precisely. When building a new task, benchmarking existing methods (SOTAs) is a common practice. But due to the new tasks and corresponding new challenges, most SOTAs perform significantly worse and thus easy to be beaten. Admitted that you built a framework for your dataset, but I do not value the proposed method. It is an A + B one.

---

> > > > > ### Author Response · Authors · 2023-08-22
> > > > >
> > > > > Thanks for your comments.
> > > > >
> > > > > Here, we suggest the **size** issue is because you mentioned your concern about our dataset. For other issues like novelty and experiments, we have listed our perspectives to facilitate respected reviewers’ and ACs' final decision. Besides, we think the effort to construct a new task and dataset should not be overlooked and is not always consistent (e.g. three months). We think our efforts in introducing the ReID-AW task to the ReID community are not negligible.
> > > > >
> > > > > Finally, we would like to extend our gratitude again.  Although we may have some inconsistent perspectives, we appreciate all your comments and will take them to improve our work.

---

> > > ### Comment · Reviewer_5HdD · 2023-08-22
> > > **Building useful and large-scale dataset is not easy**
> > >
> > > I agree that it is very difficult to improve performance on generalized large-scale benchmarks like ImageNet or CoCo. However, it is not difficult to improve performance on sub-domains such as person reid and sub-directions under it such as occlusion and cross-modality.
> > >
> > > In addition, I likewise have three pieces of work on building datasets, (one accepted, one not accepted at first, then resubmitted to IEEE Trans journal and eventually accepted, and one in review). These efforts take from three to six months, especially with only one or two people constructing these datasets.

---

> > > > ### Author Response · Authors · 2023-08-22
> > > >
> > > > Thank you for your support. We will continue to refine our dataset and the ReID-AW setting to increase its research and practical value.

---

> ### Author Response · Authors · 2023-08-22
>
> Thanks to all the respected reviewers for recognizing the value and effort of this work. With the discussion, we have presented our perspectives. Here, we still think the size should not be regard as the sole criterion to evaluate our dataset and our effort to consturct it is also not negligible. Nonetheless, we value the comments from all the reviewers and await the final decision.

---

### Decision · Program_Chairs · 2023-09-21

**Decision:**

Accept (poster)

**Comment:**

This paper has 7 (accept) -4 (borderline reject)-5 (borderline accept)-2 (strong reject) ratings. Critiques from reviewers are mainly about the novelty and limited improvement. On the other hand, reviewers also acknowledge the contribution of the dataset, the proposal of a new re-id setting and that the proposed method is tailored for this problem.

The AC read the paper and all author-reviewer correspondances and tends to agree that the pros outweight cons. The AC thinks the community would benefit from the research outcome of this paper not only in the new re-id application, but also in regards to how the proposed cross-species re-id is tentatively addressed. Regarding novelty, authors made a fair point:

*Our insights: (1) the utilization of category-adaptive features (2) the incorporation of text-guided representations to integrate pre-existing category-related knowledge from large language models, seem acknowledged.*

*These two insights we believe are novel in the ReID community and could be extended to analogous cross-category instance-level retrieval scenarios. The primary divergence in our perspectives seems to be the novel degree to these insights.*

The AC tends to agree with this argument.

That said, in regard to limited improvement, authors are strongly suggested to provide confidence interval or statistical significance tests, to demosntrate that the improvement is statistically significant. Moreover, the authors are suggested to move the attention map figure and dataset statistics table from supplementary materials to the main paper, because they are quite important.

In addition, the SAC found some papers that also released multi-species animal re-id datasets. So the authors are suggested to cite the relevant ones and make more fair claims around whether being the first to release a multi-species RE-ID dataset.

A Benchmark Database for Animal Re-Identification and Tracking (https://ieeexplore.ieee.org/stamp/stamp.jsp?tp=&arnumber=10052988)
\
https://www.sciencedirect.com/science/article/pii/S1574013720303890
\
https://sites.google.com/view/wacv2020animalreid/home

Authors are asked to search the network for multi-species RE-ID animal datasets and discuss briefly in the paper how their dataset differs from each existing dataset.

After extensive discussions between AC and SAC, this paper is recommended for accept.